# PRIVACY-PRESERVING VISION TRANSFORMER ON PERMUTATION-ENCRYPTED IMAGES

## ABSTRACT

Massive human-related data is collected to train neural networks for computer vision tasks. Potential incidents, such as data leakages, expose significant privacy risks to applications. In this paper, we propose an efficient privacy-preserving learning paradigm, where images are first encrypted via one of the two encryption strategies: (1) random shuffling to a set of equally-sized patches and (2) mixing-up sub-patches. Then, a permutation-equivariant vision transformer is designed to learn on the encrypted images for vision tasks, including image classification and object detection. Extensive experiments on ImageNet and COCO show that the proposed paradigm achieves comparable accuracy with the competitive methods. Moreover, decrypting the encrypted images is solving an NP-hard jigsaw puzzle or an ill-posed inverse problem, which is empirically shown intractable to be recovered by the powerful vision transformer-based attackers. We thus show that the proposed paradigm can destroy human-recognizable contents while preserving machine-learnable information. Code will be released publicly.

## 1  INTRODUCTION

Deep models trained on massive human-related data have been dominating many computer vision tasks, e.g., image classification He et al. (2016), face recognition Li et al. (2021), etc. However, most existing approaches are built upon images that can be recognized by human eyes, leading to the risk of privacy leaks since visually perceptible images containing faces or places may reveal privacy-sensitive information. This could raise concerns about privacy breaches, thus limiting the deployment of deep models in privacy-sensitive/security-critical application scenarios or increasing people's doubts about using deep models deployed in cloud environments.

To address the increasing privacy concerns, researchers have integrated privacy protection strategies into all phases of machine learning, such as data preparation, model training and evaluation, model deployment, and model inference Xu et al. (2021). The emerging federated learning allows multi participants to jointly train a machine learning model while preserving their private data from being exposed Liu et al. (2022). However, attackers can still recover images with high accuracy from the leaked gradients Hatamizadeh et al. (2022) or confidence information Fredrikson et al. (2015). To protect privacy-sensitive data in a confidential level, directly learning and inferencing on encrypted data is emerging as a promising direction. Unfortunately, two huge complications remain Karthik et al. (2019): (1) the encryption methods themselves, such as fully homomorphic encryption, have a very high computation complexity and (2) training deep models in the encrypted domain is an extremely challenging task due to the need for calculations in the ciphertext space.

Recent studies in the natural language processing field suggest that higher-order co-occurrence statistics of words play a major role in masked language models like BERT Sinha et al. (2021). Moreover, it has been shown that the word order contains surprisingly little information compared to that contained in the bag of words, since the understanding of syntax and the compressed world knowledge held by large models (e.g. BERT and GPT-2) are capable to infer the word order Malkin et al. (2021). Due to the property of attention operation, when removing positional encoding, Vision Transformer (ViT) Dosovitskiy et al. (2020) is permutation-equivariant w.r.t. its attentive tokens. As evaluated by our experiments, removing the positional embedding from ViT only leads to a moderate performance drop (%3.1, please see Table 1). Such a phenomenon inspires us to explore permutation-based encryption strategies.

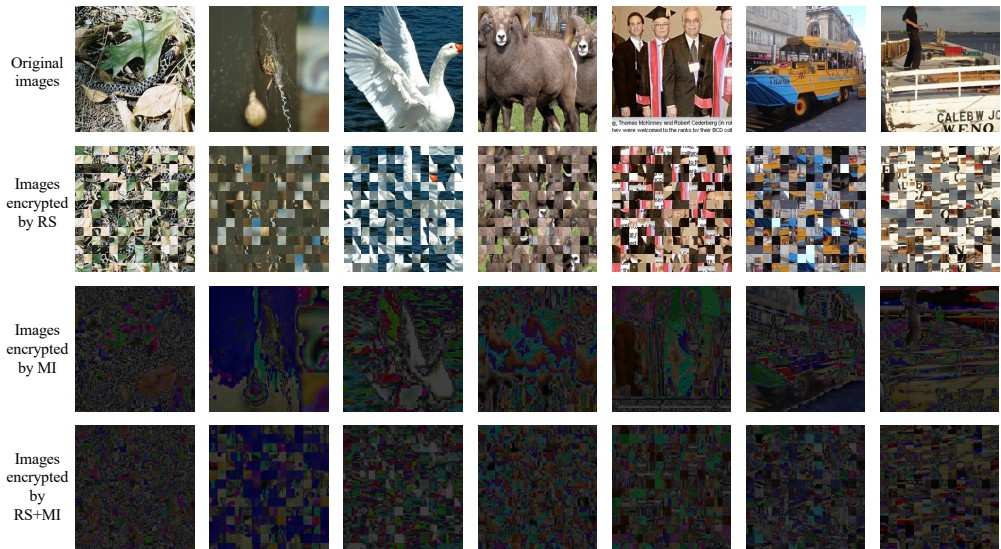

Figure 1: Illustration of images encrypted by random shuffling (RS), mixing-up (MI), and their combination. The visual contents of encrypted images are near-completely protected from recognizing by human eyes.

To maximize the usability of the resulting paradigm, the following two requirements need to be satisfied. Firstly, the encryption process should preserve the machine-learnable information of inputs. Compared with existing deep models applied on the non-encrypted images, the performance drop is expected to be insignificant and acceptable, and thus make it possible to replace existing deep models on a large scale for privacy-sensitive circumstances. Secondly, the human-recognizable contents should be largely interfered, and the decryption algorithm is expected to have a very high complexity or an unaffordable cost. In addition, it would be better if the encryption algorithm is decoupled from the models to be trained, as this would allow the encryption of image contents in scenarios where only limited computing resources are available.

To this end, we propose an efficient privacy-preserving learning paradigm. The key insight of our paradigm is two-fold: (1) designing encryption strategies based on permutation-equivariance and (2) making part of or the whole network permutation-equivariant, which allows it to learn on the encrypted images. Two strategies are proposed to encrypt images: random shuffling (RS) images to a set of equally-sized patches and mixing-up (MI) sub-patches; see Figure 1. Decrypting an image encrypted by RS is solving a jigsaw puzzle problem, which can incur a large computational overhead since the problem to be solved is an NP-hard one Demaine & Demaine (2007). Decrypting an image encrypted by MI is solving an ill-posed inverse problem, which could be hard to solve due to the difficulty in modelling the sub-patch distribution. We indicate that both kinds of encrypted images are still machine-learnable, by further designing architectures PEViT and PEYOLOS, based on ViT and YOLOS Fang et al. (2021) for the image classification and object detection tasks, respectively; see Figure 2. Specifically, our main contributions are summarized as follows:

- We propose an efficient privacy-preserving learning paradigm that can destroy human-recognizable contents while preserving machine-learnable information. The paradigm adopts a decoupled encryption process that utilizes the permutation-equivariance property so as to be still learnable for networks that are (partially) permutation-equivariant.
- RS is tailored for the standard image classification with vision transformers. By substituting reference-based positional encoding for the original one, the network is capable of learning on images encrypted by RS.
- Another hallmark of our paradigm is that by further designing MI, the paradigm is extensible to position-sensitive tasks, such as object detection, for which we adapt the way that image patches are mapped to make the network partially permutation-equivariant.
- Extensive attack experiments show the security of our encryption strategies. Comparison results on large-scale benchmarks show that both PEViT and PEYOLOS achieve promising performance even with highly encrypted images as input. We thus show that the proposed paradigm can destroy human-recognizable contents while preserving machine-learnable information.

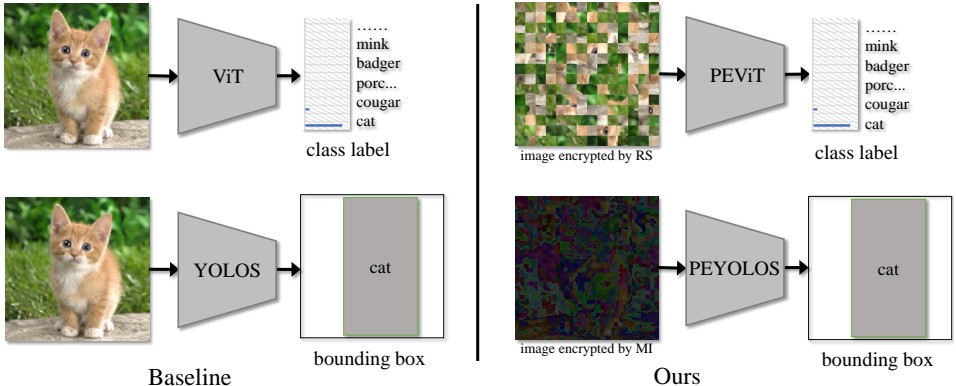

Figure 2: Images encrypted by RS and MI are still machine-learnable, by further designing architectures PEViT and PEYOLOS, based on ViT and YOLOS.

## 2 RELATED WORK

**Vision transformers.** Self-attention based Transformer Vaswani et al. (2017) has achieved great success in natural language processing. To make Transformer suitable for image classification, the pioneering work of ViT Dosovitskiy et al. (2020) directly tokenizes and flattens 2D images into a sequence of tokens. Since then, researchers have been working on improving Vision Transformers and examples include DeiT Touvron et al. (2020), T2T-ViT Yuan et al. (2021), PVT Wang et al. (2021), and Swin-Transformer Liu et al. (2021). In addition, the intriguing properties of ViT are investigated in Naseer et al. (2021).

**Jigsaw puzzle solver.** The goal of a jigsaw puzzle solver is to reconstruct an original image from its shuffled patches. Since this problem is NP-hard Demaine & Demaine (2007), solving jigsaw puzzles of non-trivial size is impossible. Most of the existing works in computer vision focus on the jigsaw puzzle problem composed of equally-sized image patches and examples include the greedy algorithm proposed in Cho et al. (2010), the particle filter-based algorithm proposed in Yang et al. (2011), the fully-automatic solver proposed in Pomeranz et al. (2011), and the genetic-based solver proposed in Sholomon et al. (2013).

**Privacy-preserving machine learning.** The aim of privacy-preserving machine learning is to integrate privacy-preserving techniques into the machine learning pipeline. According to the phases of privacy integration, existing methods can be basically divided into four categories: data preparation, model training and evaluation, model deployment, and model inference Xu et al. (2021). Federated learning allows multiple participants to jointly train a machine learning model while preserving their private data from being exposed Liu et al. (2022). However, the leakage of gradients Hatamizadeh et al. (2022) or confidence information Fredrikson et al. (2015) can be utilized to recover original images such as human faces. From the perspective of data, encrypting data and then learning and inferencing on encrypted data can provide a strong privacy guarantee, called confidential-level privacy, which receives increasing attention recently. Both homomorphic encryption and functional encryption have been employed to encrypt data due to their nature of allowing computation over encrypted data, which enables machine learning on encrypted data Xu et al. (2019); Karthik et al. (2019). However, scaling them to deep networks and large datasets still faces extreme difficulties due to the high computational complexity involved. The block-wise pixel shuffling encryption strategy Tanaka (2018); Madono et al. (2020) and the pixel-based image encryption strategy Sirichotedumrong et al. (2019) have low computational complexity, but scaling them to position-sensitive tasks faces great challenges. In contrast, our permutation-based encryption strategies are simple, efficient and easy to implement, and can be applied to position-sensitive tasks.

## 3 METHOD

In this section, we propose an efficient privacy-preserving learning paradigm that can destroy human-recognizable contents while preserving machine-learnable information. We first provide the permutation-based encryption strategies, in Sec. 3.1. Followed by the designed ViT adaptions which are (partially) permutation-equivariant so as to handle the encrypted inputs, in Sec. 3.3 and Sec. 3.4.

### 3.1 Destroying Human-Recognizable Contents

We first consider typical vision tasks which are not quite position-sensitive, such as image classification that predicts the global category. Here, *Random Shuffling (RS)* images to a set of equally-sized image patches can destroy human-recognizable contents. This shuffle-encrypted process is simple, easy to implement, and is decoupled from the network optimization. Under the circumstance, decrypting an image is solving a jigsaw puzzle problem, which can incur a large computational overhead since the problem to be solved is an NP-hard one Demaine & Demaine (2007). In particular, the dimension of the key space when applying the shuffle-encrypted strategy is the number of puzzle permutations. For an image with $N$ patches, the dimension of the key space is given by $K_S = N!$. For example, a $7 \times 7$ puzzle has $49! \approx 6 \times 10^{62}$ possible permutations. Based on this, it is easy to further increase the complexity of decrypting an image, by reducing the patch size of puzzles or increasing the resolution of the image. Besides, the complexity of decrypting an image can be further increased by dropping some image patches, as experimentally evaluated in Figure 5 and Figure 4.

However, always shuffling image patches is prone to underestimating the position information. To further make our paradigm applicable to position-sensitive tasks like object detection, where precise positions of bounding boxes are predicted, we design another encryption strategy named *Mixing (MI)*. Specifically, MI mixes sub-patches of image patches to destroy human-recognizable contents while preserving the position information, so that the encrypted data can be learned by networks that are partially permutation-equivariant. Assuming that there are $M$ sub-patches extracted from an image patch $x^p$, the mixing-encrypted strategy can be formulated as follows,

$$\mathbf{x}_S^p = \frac{1}{M} \sum_{i=1}^{M} \mathbf{s}_i^p , \tag{1}$$

where $\mathbf{s}_i^p$ denotes the $i$-th sub-patch of $x^p$. Since the sum function is permutation-equivariant, MI makes an encrypted image partially permutation-equivariant to its sub-patches. With the MI encryption process, decrypting a patch is solving the following ill-posed inverse problem,

$$\operatorname*{arg\,min}_{\mathbf{s}_1^p, \cdots, \mathbf{s}_M^p} \| \mathbf{x}_S^p - \sum_{i=1}^{M} \mathbf{s}_i^p \|^2 , \tag{2}$$

Both modeling the sub-patch distribution and restoring the sub-patch order make decrypting an patch a great challenge. Decrypting an image of $N$ patches magnifies this challenge by a factor of $N$.

### 3.2 Building Blocks of ViT

In this part, we analyze how the change of input permutation affects each component of ViT.

**Self-attention.** The attention mechanism is a function that outputs the weighted sum of a set of $k$ *value* vectors (packed into $V \in \mathbb{R}^{k \times d}$). The $k$ weights are obtained by calculating the similarity between a *query* vector $q \in \mathbb{R}^d$ and a set of $k$ *key* vectors (packed into $K \in \mathbb{R}^{k \times d}$) using inner products, which are then scaled and normalized with a softmax function. For a sequence of $N$ query vectors (packed into $Q \in \mathbb{R}^{N \times d}$), the output matrix $O$ (of size $N \times d$) can be computed by,

$$O = \mathrm{Attention}(Q, K, V) = \mathrm{Softmax}(QK^\top / \sqrt{d})V, \tag{3}$$

where the Softmax function is applied to the input matrix by rows.

In self-attention, *query*, *key*, and *value* matrices are computed from the same sequence of $N$ input vectors (packed into $X \in \mathbb{R}^{N \times d}$) using linear transformations: $Q = XW^\mathrm{Q}$, $K = XW^\mathrm{K}$, $V = XW^\mathrm{V}$. Since the order of $Q$, $K$, and $V$ is co-variant with that of $X$, the permutation of the input of self-attention permutes the output.

**Multi-head self-attention.** Multi-head Self-Attention (MSA) Vaswani et al. (2017) consists of $h$ self-attention layers, each of which outputs a matrix of size $N \times d$. These $h$ output matrices are then rearranged into a $N \times dh$ matrix that is reprojected by a linear layer into $N \times d$,

$$\mathrm{MSA} = \mathrm{Concat}(\mathrm{head}_1, \cdots, \mathrm{head}_h)W^\mathrm{O}, \tag{4}$$

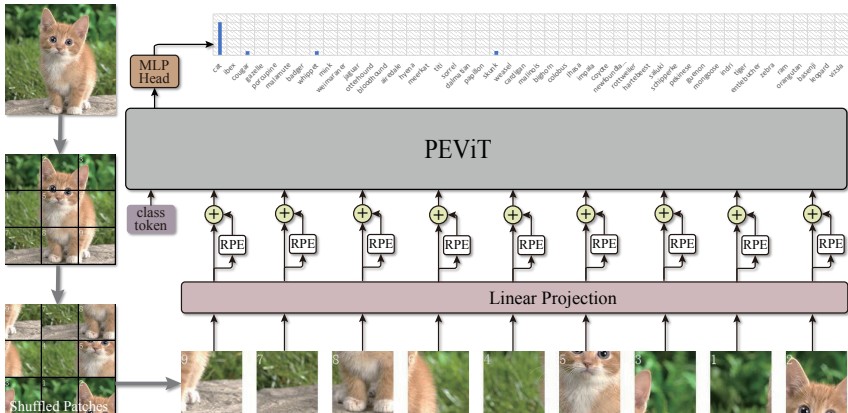

Figure 3: Architecture overview of PEVIT with RPE.

where $\text{head}_i = \text{Attention}(Q_i, K_i, V_i)$, $Q_i = XW_i^{\text{Q}}$, $K_i = XW_i^{\text{K}}$, $V_i = XW_i^{\text{V}}$, and $W^{\text{O}} \in \mathbb{R}^{dh \times d}$. The order of each head is co-variant with that of $X$. Since the Concat operation concatenates the vectors from different heads at the same position, it is co-variant to the order of $X$. Therefore, we conclude that the permutation of the input of MSA permutes the output.

**Layer normalization.** Layer normalization Ba et al. (2016) is applied to the last $d$ dimensions, based on which the mean and standard deviation are calculated. Therefore, the permutation of the input of layer normalization permutes the output.

**Residual connection.** The residual connection He et al. (2016) can be formulated as $\mathcal{F}(X) + X$. If the order of the nonlinear mapping $\mathcal{F}(\cdot)$ is co-variant with that of $X$, the residual connection is co-variant with that of $X$.

**Feed-forward network.** Feed-Forward Network (FFN) consists of two linear layers separated by a GELU activation Hendrycks & Gimpel (2016). The first linear layer expands the dimension from $D$ to $4D$, while the second layer reduces the dimension from $4D$ back to $D$. Since FFN is applied on the last $d$ dimensions, the permutation of the input of FFN permutes the output.

### 3.3 CLASSIFICATION ON ENCRYPTED IMAGES

As a standard method to handle images in ViT, the fixed-size input image of $H \times W \times C$ is decomposed into a batch of $N$ patches of a fixed resolution of $P \times P$, resulting in a sequence of flattened 2D patches $X^p \in \mathbb{R}^{N \times (P^2 \cdot C)}$. For example, the sequence length $N = HW/P^2$ of ViT could be 196 for image classification on the ImageNet dataset. To destroy the human-recognizable contents, we choose RS as the encryption strategy to encrypt images. The reason is two-fold: (1) The key space of an image encrypted by RS is big enough and (2) The drop in performance is insignificant. To learn on the images encrypted by RS, we design Permutation-Equivariant ViT (PEViT), defined as follows,

$$\{\mathbf{x}_1^p, \mathbf{x}_2^p, \cdots, \mathbf{x}_N^p\} \xrightarrow{\text{Shuffling}} \{\mathbf{x}_2^p, \mathbf{x}_N^p, \cdots, \mathbf{x}_1^p\} \tag{5}$$

$$\mathbf{z}_0 = [\mathbf{x}_{\text{class}}; \mathbf{x}_2^p\mathbf{E}; \mathbf{x}_N^p\mathbf{E}; \cdots; \mathbf{x}_1^p\mathbf{E}], \qquad \mathbf{E} \in \mathbb{R}^{(P^2 \cdot C) \times D} \tag{6}$$

$$\mathbf{z'}_\ell = \text{MSA}(\text{LN}(\mathbf{z}_{\ell-1})) + \mathbf{z}_{\ell-1}, \qquad \ell = 1 \ldots L \tag{7}$$

$$\mathbf{z}_\ell = \text{FFN}(\text{LN}(\mathbf{z'}_\ell)) + \mathbf{z'}_\ell, \qquad \ell = 1 \ldots L \tag{8}$$

$$\mathbf{y} = \text{LN}(\mathbf{z}_L^0). \tag{9}$$

where $\mathbf{E}$ denotes the linear projection that maps each vectorized image patch to the model dimension $D$, and $\mathbf{x}_{\text{class}}$ denotes the class token ($\mathbf{z}_0^0 = \mathbf{x}_{\text{class}}$), whose state at the output of the visual transformer ($\mathbf{z}_L^0$) serves as the image representation $\mathbf{y}$.

The differences between PEViT and vanilla ViT are two-fold: (1) Our model takes shuffled patch embeddings as input and (2) The learned positional encodings are removed. Since the permutation of the input of all the building blocks permutes the output, the order of $\mathbf{z}_L$ is co-variant with that of $\mathbf{z}_0$. It is worth noting that the class token is fixed in $\mathbf{z}_0^0$. Therefore, $\mathbf{z}_L^0$ corresponds to the image representation $\mathbf{y}$ that is equivariant to the order of patch embeddings or image patches.

When directly introducing positional encoding, the permutation-equivariance property of PEViT is destroyed. Inspired by relative encoding Shaw et al. (2018), we propose a reference-based positional embedding approach that can retain the permutation-equivariance property,

$$\mathbf{E}_{pos}(\mathbf{x}_i^p) = \text{RPE}(\mathbf{x}_i^p - \mathbf{x}^{\text{ref}}), \tag{10}$$

where $\mathbf{x}^{\text{ref}} \in \mathbb{R}^d$ denotes the learnable reference point and RPE denotes the reference-based positional encoding network that consists of two linear layers separated by a GELU activation Hendrycks & Gimpel (2016), followed by a sigmoid function. This reference-based positional embedding relies only on the learnable reference point and patch embeddings, and thus its order is co-variant with that of input vectors. When added to the patch embedding as follows,

$$\mathbf{z}_0 = [\mathbf{x}_{\text{class}}; \ \mathbf{x}_2^p\mathbf{E} + \mathbf{E}_{pos}(\mathbf{x}_2^p); \ \mathbf{x}_N^p\mathbf{E} + \mathbf{E}_{pos}(\mathbf{x}_N^p); \cdots; \ \mathbf{x}_1^p\mathbf{E} + \mathbf{E}_{pos}(\mathbf{x}_1^p)], \tag{11}$$

where the permutation-equivariance property of PEViT is retained. An illustration of PEViT with RPE is depicted in Figure 3.

## 3.4 Object Detection on Encrypted Images

YOLOS Fang et al. (2021) is an object detection model based on the vanilla ViT. The change from a ViT to a YOLOS involves two steps: (1) Dropping the class token and appending 100 randomly initialized learnable detection tokens to the input patch embeddings and (2) Replacing the image classification loss with the bipartite matching loss to perform object detection in a set prediction manner Carion et al. (2020).

Since position information plays a key role in the low-level object detection task, directly encrypting images with RS messes up the patch positions and thus leads to significant performance degradation. This brings a great challenge to destroy human-recognizable contents while preserving machine-learnable information for the object detection task. We address this challenge by adapting the way image patches are mapped. For an image patch ($\mathbf{x}_i^p$) of a fixed resolution $P \times P$, we further decompose the patch into a batch of 4 sub-patches of a fixed resolution $\frac{P}{2} \times \frac{P}{2}$. Then, these sub-patches are encrypted with MI, resulting in the encrypted patch $\mathbf{x}_i^S$. Accordingly, the input to YOLOS is adapted as follows,

$$\mathbf{z}_0 = [\mathbf{x}_1^{DET}; \cdots; \mathbf{x}_{100}^{DET}; \ \mathcal{H}(\mathbf{x}_1^S); \mathcal{H}(\mathbf{x}_2^S); \cdots; \mathcal{H}(\mathbf{x}_N^S)] + \mathbf{P}, \tag{12}$$

where $\mathcal{H}$ denotes a nonlinear mapping composed of two linear layers separated by a GELU activation Hendrycks & Gimpel (2016) and $\mathbf{P} \in \mathbb{R}^{(100+N) \times D}$ denotes the learnable positional embeddings. This adaptation makes YOLOS partially permutation-equivariant to its sub-patches. With our PEYOLOS, we can destroy human-recognizable contents while preserving machine-learnable information for the object detection task. More details can be found in the appendix.

## 3.5 Discussion on The Connection between RS and MI

RS shuffles the patch order of an image, which destroys the position configurations between patches. MI mixes up the sub-patches in a patch, which preserves the position configurations between patches, and can thus be applied to position-sensitive scenarios. Since MI encrypts images only at the sub-patch level, it is also applicable to position-insensitive scenarios such as image classification. Considering that a key patch of an image encrypted by RS may reveal information to identify an object or person, RS and MI can even be combined to further enhance the degree of data protection; see Figure 1.

## 4 Experiments

In this section, we first contrast the performance of the proposed PEViT and PEYOLOS on image classification and object detection tasks, respectively. Due to limited space, ablation studies are provided in the appendix.

### 4.1 Experimental settings

**Datasets.** For the image classification task, we benchmark the proposed PEViT on ImageNet-1K Deng et al. (2009), which contains ∼1.28M training images and 50K validation images. For the

Table 1: Comparison of different methods on ImageNet-1K classification.

| method | image size | #param. | FLOPs | ImageNet top-1 acc. |
|---|---|---|---|---|
| RegNetY-16G Radosavovic et al. (2020) | $224^2$ | 84M | 16.0G | 82.9 |
| EffNet-B3 Tan & Le (2019) | $300^2$ | 12M | 1.8G | 81.6 |
| EffNet-B4 Tan & Le (2019) | $380^2$ | 19M | 4.2G | 82.9 |
| EffNet-B5 Tan & Le (2019) | $456^2$ | 30M | 9.9G | 83.6 |
| DeiT-B Touvron et al. (2020) | $224^2$ | 86M | 17.5G | 81.8 |
| TNS-B Han et al. (2021) | $224^2$ | 66M | 14.1G | 82.8 |
| T2T-ViT-14 Yuan et al. (2021) | $224^2$ | 22M | 5.2G | 81.5 |
| T2T-ViT-24 Yuan et al. (2021) | $224^2$ | 64M | 14.1G | 82.3 |
| PVT-Large Wang et al. (2021) | $224^2$ | 61M | 9.8G | 81.7 |
| Swin-T Liu et al. (2021) | $224^2$ | 28M | 4.5G | 81.2 |
| Swin-S Liu et al. (2021) | $224^2$ | 50M | 8.7G | 83.2 |
| GG-T Yu et al. (2021) | $224^2$ | 28M | 4.5G | 82.0 |
| GG-S Yu et al. (2021) | $224^2$ | 50M | 8.7G | 83.4 |
| ViT-B/16 Dosovitskiy et al. (2020) | $384^2$ | 86M | 55.4G | 77.9 |
| ViT-L/16 Dosovitskiy et al. (2020) | $384^2$ | 307M | 190.7G | 76.5 |
| **DeiT-B** on images encrypted by RS | $224^2$ | 86M | 17.5G | 35.9 |
| **DeiT-B** on images encrypted by MI | $224^2$ | 86M | 17.5G | 78.0 |
| **PEViT-B** on images encrypted by RS | $224^2$ | 86M | 17.5G | 78.7 |
| **PEViT-B** on images encrypted by RS + MI | $224^2$ | 86M | 17.5G | 69.5 |
| **PEViT-B** with RPE on images encrypted by RS | $224^2$ | 87M | 17.6G | 79.7 |

object detection task, we benchmark the proposed PEYOLOS on COCO Lin et al. (2014), which contains 118K training, 5K validation and 20K test images.

**Implementation details.** [1] We implement the proposed PEViT based on the Timm library Wightman (2019). We adopt the default hyper-parameters of the DeiT training scheme Touvron et al. (2020) except setting the batch size to 192 per GPU, where 8 NVIDIA A100 GPUs are used for training. It is worth noting PEViT (w.o RPE) is equivalent to removing positional embeddings from DeiT. We implement the proposed PEYOLOS based on the publicly released code in Fang et al. (2021) and change the way image patches are mapped.

**Baseline.** We propose an efficient privacy-preserving learning paradigm with the aim of destroying human-recognizable contents while preserving machine-learnable information. The proposed PEViT and PEYOLOS are inherited from DeiT Touvron et al. (2020) and YOLOS Fang et al. (2021) respectively, which are thus selected as baselines. It is worth noting that both PEViT and PEYOLOS are not designed to be high-performance models that beats state-of-the-art image classification and object detection models, but to unveil that destroying human-recognizable contents while preserving machine-learnable information is feasible.

**Measurement of privacy protection.** As shown in Figure 1, the visual contents of encrypted images are nearly-completely protected from recognizing by human eyes. To measure the strength of privacy protection, we try to restore the original images with various attack algorithms, including puzzle solver attacker, gradient leakage attacker, and reconstruct attacker. Then, the quality of restored images can reflect the strength of privacy protection. We only provide attack experiments from puzzle solver attacker in the main text and more attack experiments can be found in the appendix.

## 4.2 IMAGENET-1K CLASSIFICATION

The summary of the image classification results is shown in Table 1. It can be observed that (1) Our performance drop is only 3.9% compared with the state-of-the-art CNN-based method EffNet-B5, (2) Our performance drop is only 2.1% compared with the baseline method DeiT-B, and (3) MI is applicable to image classification. It is worth noting that although the performance of PEViT-B is not state-of-the-art, it is applied on encrypted images where the visual contents of images cannot

---

[1] The pseudocode of RS and MI in a PyTorch-like style are shown in the appendix.

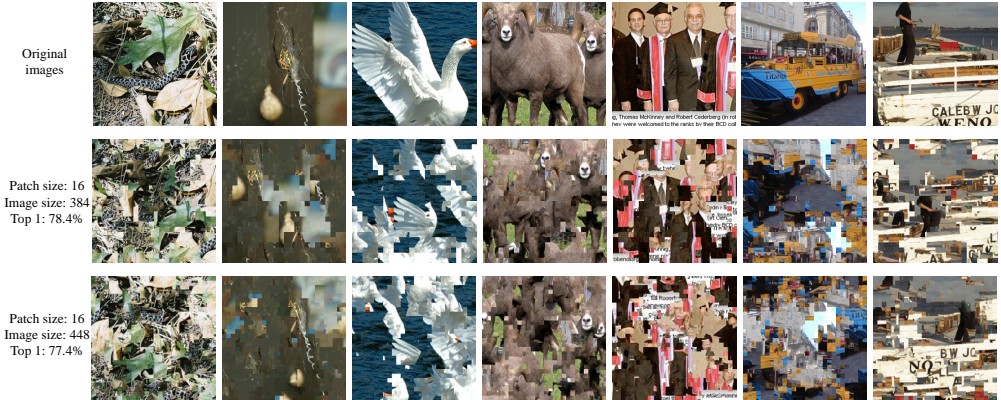

Figure 4: Reconstructed images by the jigsaw puzzle solver proposed in Paikin & Tal (2015). Here, the effect of image size on image reconstruction quality and classification performance is investigated.

Table 2: Object detection performance on the COCO test2017 dataset. FPS is measured with batch size 1 on a single 1080Ti GPU.

| method | backbone | size | AP | params. | FLOPs | FPS |
|---|---|---|---|---|---|---|
| YOLOS-Ti | DeiT-Ti | $512 \times *$ | 28.7 | 6.5M | 18.8G | 60 |
| Deformable DETR | FBNet-V3 | $800 \times *$ | 27.9 | 12.2M | 12.3G | 35 |
| YOLOSv4-Tiny | COSA | $416 \times *$ | 21.7 | 6.1M | 7.0G | 371 |
| CenterNet | ResNet-18 | $512 \times *$ | 28.1 | - | - | 129 |
| **PEYOLOS** on images encrypted by MI | DeiT-Ti | $512 \times *$ | 25.3 | 7.1M | 19.0G | 58 |
| DETR | ResNet-18 | $800 \times *$ | 36.9 | 29M | 129G | 7.4 |
| YOLOS-S | DeiT-S | $800 \times *$ | 36.1 | 31M | 194G | 5.7 |
| **PEYOLOS** on on images encrypted by MI | DeiT-S | $800 \times *$ | 32.9 | 31.6M | 194.9G | 5.6 |
| DETR | ResNet-101 | $800 \times *$ | 42.5 | 60M | 253G | 5.3 |
| YOLOS-B | DeiT-B | $800 \times *$ | 42.0 | 127M | 538G | 2.7 |
| **PEYOLOS** on on images encrypted by MI | DeiT-B | $800 \times *$ | 39.5 | 128.2M | 539.7G | 2.5 |

be recognized by human eyes, while the comparison methods cannot. Moreover, RS and MI can be combined to further improve data security but at the expense of performance. Therefore, we conclude that PEViT-B achieves a trade-off between performance and visual content protection.

## 4.3 COCO OBJECT DETECTION

The summary of the object detection results is shown in Table 2, in which we compare PEYOLOS with the competitive methods that contains roughly equal parameters. It can be observed that PEYOLOS is comparable to these methods. In particular, our performance drop is only ∼3.0% compared with the baseline method YOLOS, and only 2.8% compared with the CNN-based method CenterNet. Similar to PEViT, although PEYOLOS does not outperform the state-of-the-art methods, it is currently the unique model to achieve a trade-off between performance and visual content protection for object detection.

## 4.4 PUZZLE SOLVER ATTACKER

Jigsaw puzzle solver, i.e., reconstructing the image from the set of shuffled patches, can be used to attack the images encrypted with MS. Since this problem is NP-hard Demaine & Demaine (2007), solving jigsaw puzzles of non-trivial size is impossible. Most of the existing works in computer vision focus on the jigsaw puzzle problem composed of equally-sized image patches Cho et al. (2010); Pomeranz et al. (2011); Sholomon et al. (2013); Paikin & Tal (2015), in which only pixels that are no more than two pixels away from the boundary of a piece are utilized. Therefore, with adaptations on how the image is decomposed, these methods might fail.

We choose the solver proposed in Paikin & Tal (2015) to inveterate the effect of attacks. The reason is two-fold: (1) It is a fast, fully-automatic, and general solver, which assumes no prior knowledge

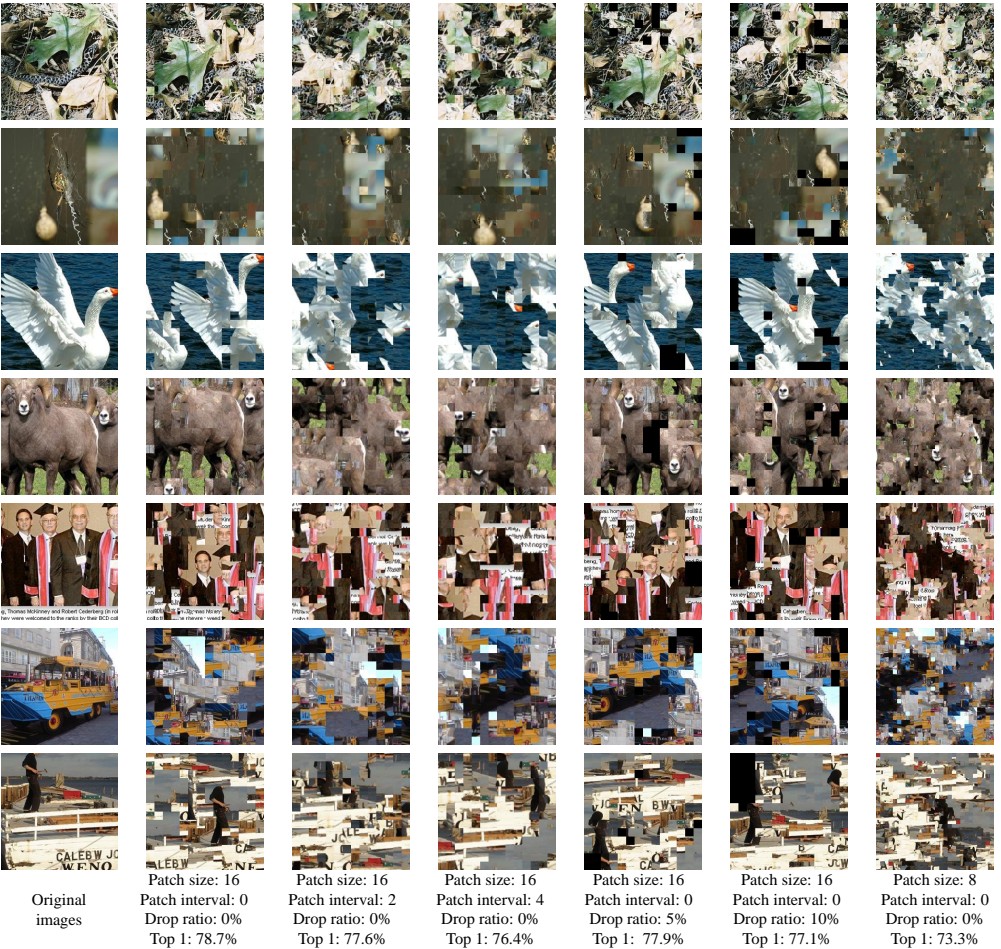

Figure 5: Reconstructed images by the jigsaw puzzle solver proposed in Paikin & Tal (2015), where the default image size is $224 \times 224$. Here, the effect of patch size, patch interval, and patch drop ratio on image reconstruction quality and classification performance is investigated.

about the original image and (2) It can handle puzzles with missing pieces. We investigate the effect of patch size, patch interval, patch drop ratio, and image size on image reconstruction quality and classification performance. The results are shown in Figure 5 and Figure 4. We observe that increasing the number of patches (reducing the patch size or increasing the image size) or the patch interval can enhance the strength of privacy protection at the cost of performance. *More importantly, even if 10% of the image blocks are dropped, the performance is only reduced by 1.6%, which allows users to drop patches containing sensitive information.*

## 5   CONCLUSION

In this paper, we propose an efficient privacy-preserving learning paradigm that can destroy human-recognizable contents while preserving machine-learnable information. The key insight of our paradigm is to decouple the encryption algorithm from the network optimization via permutation-equivariance. Two encryption strategies are proposed to encrypt images: random shuffling to a set of equally-sized image patches and mixing image patches that are permutation-equivariant. By adapting ViT and YOLOS with minimal adaptations, they can be made (partially) permutation-equivariant and are able to handle encrypted images. Extensive experiments on ImageNet and COCO show that the proposed paradigm achieves comparable accuracy with the competitive methods, meanwhile destroying human-recognizable contents.

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

# A   APPENDIX

The supplementary materials collect the source code package and the appendices on additional experiment results.

# B   IMAGE CLASSIFICATION

**Attention map.**   We refer to "discriminative image patches" as the image patches that contain contents of the target category. To give an intuitive understanding of whether PEViT-B and PEViT-B with RPE can locate discriminative image patches, we provide in Figure 6 the visualization results of attention maps from different layers of PEViT-B, PEViT-B with RPE, and DeiT-B. It can be observed that (1) The attention of the class token of PEViT-B, PEViT-B with RPE, and DeiT-B becomes more and more concentrated as the number of layers increases and (2) The discriminative image patches of PEViT-B, PEViT-B with RPE, and DeiT-B are only partially attended by the class token at the lower layers. The reason for this might be that the context between image patches plays the key role in the image classification task, or the success of PEViT-B, PEViT-B with RPE, and DeiT-B is largely due to their ability to model high-order co-occurrence statistics between image patches. Therefore, dropping out some patches has no significant impact on performance; see Figure 5.

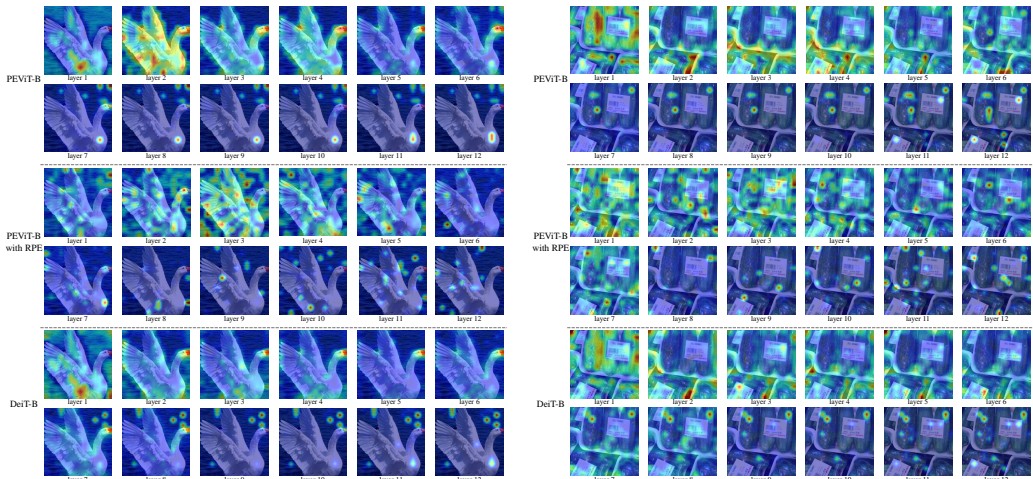

Figure 6: Attention map visualization results of the *class token* in different layers of PEViT-B, PEViT-B with RPE, and DeiT-B for comparison.

**Learned embedding filter visualization.**   To further understand the distinctions between PEViT-B, PEViT-B with RPE, and DeiT-B, we provide in Figure 7 the top 48 principal components of the learned embedding filters. Each filter resembles plausible basis functions for a low-dimensional representation of the fine structure within each patch Dosovitskiy et al. (2020). It can be observed that the filters of DeiT-B have more structure than that of both PEViT-B and PEViT-B with RPE. This means that DeiT-B focuses on specific structures that contribute to classification, while PEViT-B and PEViT-B with RPE tend to focus more on the co-occurrence statistics between image patches. Figure 8 shows the attention map of PEViT-B, PEViT-B with RPE, and DeiT-B from the first layer. The distribution of attention of both PEViT-B and PEViT-B with RPE is more dispersed than that of DeiT-B. The reason for this might be that the co-occurrence statistics are more important in PEViT-B and PEViT-B with RPE due to their lack of perception of structural information contained in images. To some extent, this phenomenon is similar to the conclusion in the natural language processing field that higher-order co-occurrence statistics of words play a major role in learning Sinha et al. (2021); Malkin et al. (2021), as discussed in Sec. 1 of our main paper.

**Feature visualization.**   To give an intuitive understanding of the distinctions between PEViT-B, PEViT-B with RPE, and DeiT-B, we provide in Figure 9 the t-SNE feature visualization results of PEViT, PEViT-B with RPE, and DeiT-B. It is worth noting that both PEViT-B and PEViT-B with

RPE can be generalized on encrypted images that are randomly shuffled. Interestingly, similar to DeiT-B, our PEViT-B and PEViT-B with RPE can still cluster the features belonging to the same class together while keeping the features belonging to different classes farther apart. This indicates the effectiveness of our permutation-equivariant designs on handling highly encrypted data.

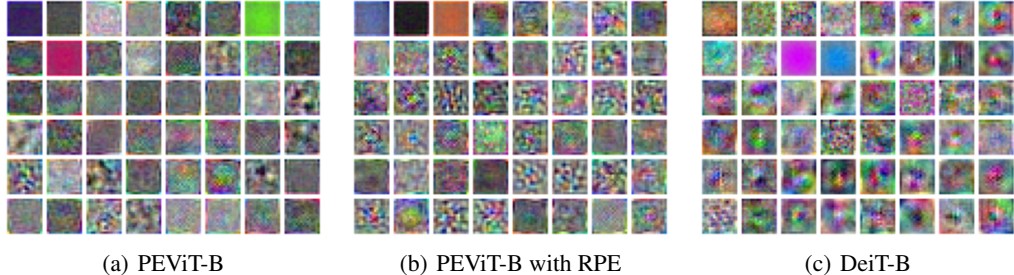

(a) PEViT-B              (b) PEViT-B with RPE              (c) DeiT-B

Figure 7: Learned embedding filter visualization. The top 48 principal components of the learned embedding filters are shown.

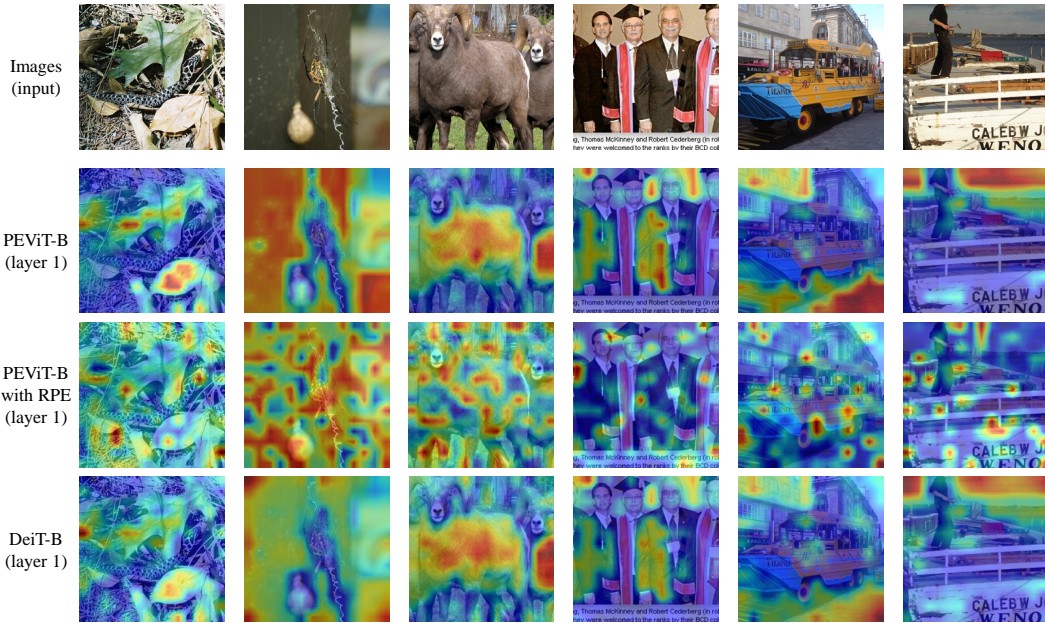

Figure 8: Attention map visualization results from PEViT-B, PEViT-B with RPE, and DeiT-B. The attention maps of the class token in the first layer are shown.

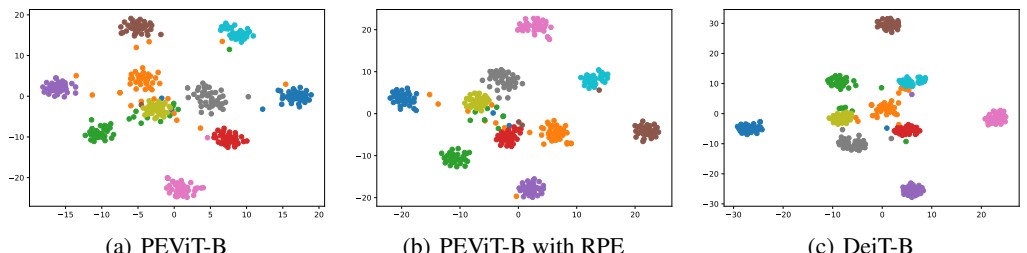

(a) PEViT-B              (b) PEViT-B with RPE              (c) DeiT-B

Figure 9: t-SNE feature visualization of PEViT-B, PEViT-B with RPE, and DeiT-B. Features from 10 randomly sampled classes in the validation set are visualized.

**Impact of RPE.** The results of PEViT with and without RPE are shown in Table 3. It can be observed that RPE can boost the performance of PEViT. The motivation for designing RPE is as

follows. In PEViT, the positional encoding is removed due to its non-permutation-equivariant property. Through RPE, we illustrate that introducing positional embedding while retaining permutation-equivariant property is feasible, which might inspire other better designs of positional embedding that is permutation-equivariant.

Table 3: Impact of RPE on image classification performance on ImageNet.

| Method | ImageNet top-1 acc. |
|---|---|
| PEViT without RPE | 78.7 |
| PEViT with RPE | 79.7 |

## C  OBJECT DETECTION

**Pipeline.** The pipeline of PEYOLOS is shown in Figure 10. The input image of $H \times W \times C$ is decomposed into a batch of $N$ patches with a fixed resolution of $P \times P$. Then, for an image patch, we further decompose the patch into a batch of 4 sub-patches with a fixed resolution $\frac{P}{2} \times \frac{P}{2}$. Finally, these sub-patches are encrypted with mixing-up (MI), resulting in a highly encrypted image that is not human-recognizable and is tough to be decrypted. PEYOLOS takes as input the encrypted images and outputs class and bounding box predictions for the original images.

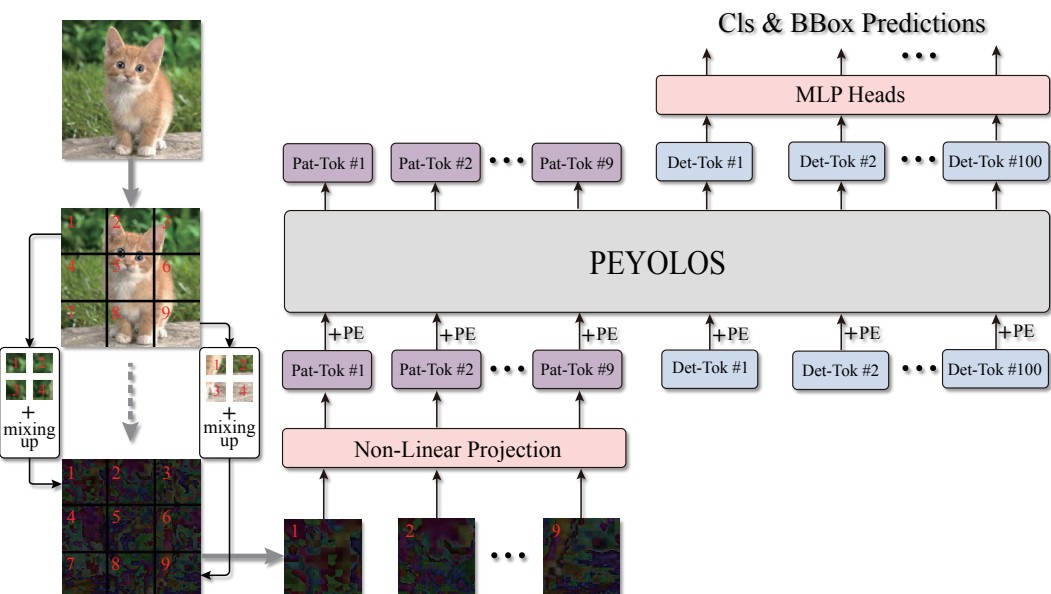

Figure 10: Architecture overview of PEYOLOS.

**Comparison results.** The summary of more object detection results is shown in Table 4, in which we compare PEYOLOS with the competitive methods that contain comparable parameters. It can be observed that (1) When using the backbone DeiT-Ti, our performance drop is only 3.4% compared with the baseline method YOLOS, (2) When using the backbone DeiT-S, our performance drop is only 3.2% compared with the baseline method YOLOS, and (3) When using the backbone DeiT-B, our performance drop is only 2.5% compared with the baseline method YOLOS. Similar to PEViT, although PEYOLOS does not outperform the state-of-the-art methods, it is currently the unique model to achieve a trade-off between performance and visual content protection for object detection; see Figure 11.

Table 4: Object detection performance on the COCO test2017 dataset. FPS is measured with batch size 1 on a single 1080Ti GPU.

| method | backbone | size | AP | params. | FLOPs | FPS |
|---|---|---|---|---|---|---|
| YOLOS-Ti | DeiT-Ti | $512 \times *$ | 28.7 | 6.5M | 18.8G | 60 |
| Deformable DETR | FBNet-V3 | $800 \times *$ | 27.9 | 12.2M | 12.3G | 35 |
| YOLOSv4-Tiny | COSA | $416 \times *$ | 21.7 | 6.1M | 7.0G | 371 |
| CenterNet | ResNet-18 | $512 \times *$ | 28.1 | - | - | 129 |
| **PEYOLOS** on images encrypted by MI | DeiT-Ti | $512 \times *$ | 25.3 | 7.1M | 19.0G | 58 |
| DETR | ResNet-18 | $800 \times *$ | 36.9 | 29M | 129G | 7.4 |
| YOLOS-S | DeiT-S | $800 \times *$ | 36.1 | 31M | 194G | 5.7 |
| **PEYOLOS** on images encrypted by MI | DeiT-S | $800 \times *$ | 32.9 | 31.6M | 194.9G | 5.6 |
| DETR | ResNet-101 | $800 \times *$ | 42.5 | 60M | 253G | 5.3 |
| YOLOS-B | DeiT-B | $800 \times *$ | 42.0 | 127M | 538G | 2.7 |
| **PEYOLOS** on images encrypted by MI | DeiT-B | $800 \times *$ | 39.5 | 128.2M | 539.7G | 2.5 |

It is worth noting that MI is not limited to YOLOS. With minimal adaptations, other object detection frameworks based on plain Vision Transformer (ViT) can also be adapted to work on images encrypted by MI. Recently, it has been shown that, with ViT backbones pre-trained as Masked Autoencoders, ViTDet Li et al. (2022) can compete with the previous leading methods that were all based on hierarchical backbones. By adapting the way the image patches are mapped like PEYOLOS, PEViTDet can be readily obtained, achieving an AP of 41.2.

**Qualitative results on COCO.** To show the difference between PEYOLOS and YOLOS, we provide in Figure 11 the qualitative results on COCO of both PEYOLOS and YOLOS. The main difference between PEYOLOS and YOLOS is that PEYOLOS takes encrypted images as input while YOLOS take unencrypted images as input. Compared with YOLOS, the performance degradation of PEYOLOS is only $\sim 3\%$. Based on these observations, we conclude that the proposed learning paradigm can destroy human-recognizable contents while preserving machine-learnable information.

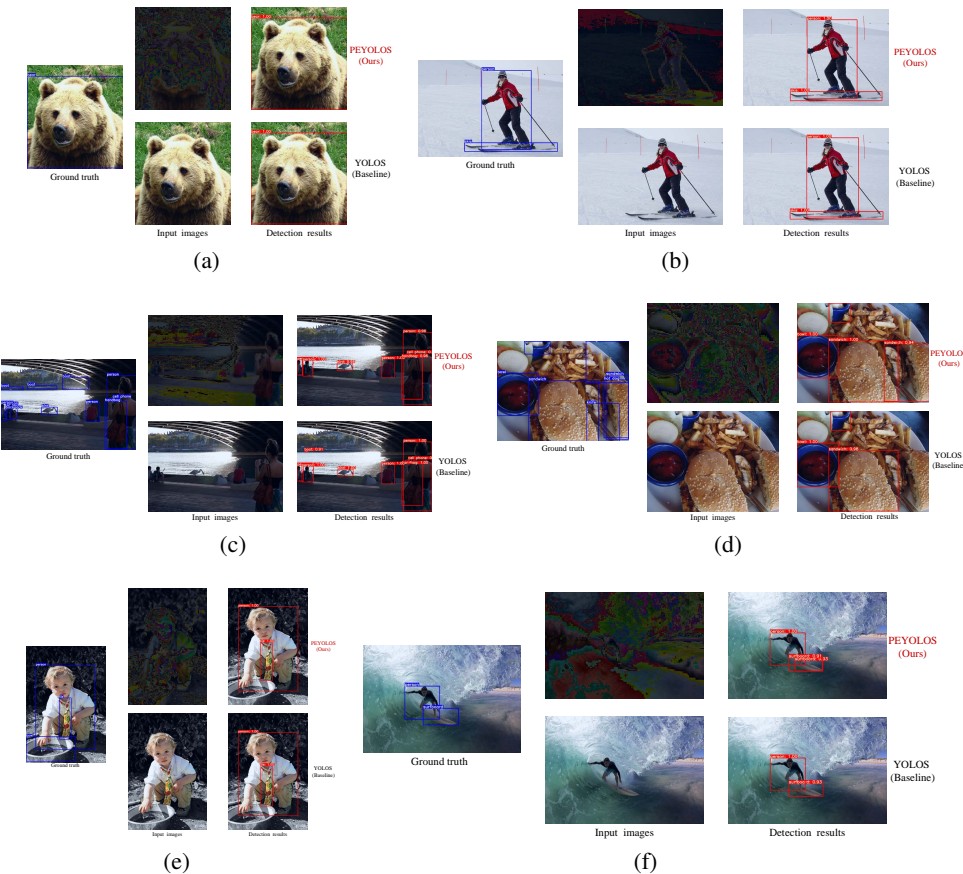

Figure 11: Qualitative results on COCO with the backbone Deit-S. The main difference between PEYOLOS and YOLOS is that PEYOLOS takes highly encryted images which are not human-recognizable and are tough to be decrypted as input while YOLOS take unencrypted images as input.

**Attention maps of detection tokens.** We inspect the attention maps of detection tokens that are related to the predictions. The experiments are conducted based on PEYOLOS and YOLOS, and the visualization results of the last layer are shown in Figure 12 and Figure 13. It can be observed that (1) For both PEYOLOS and YOLOS, different attention heads focus on different patterns at different locations, (2) The corresponding detection tokens as well as the attention map patterns are usually different for PEYOLOS and YOLOS, and (3) Some visualizations are interpretable while others are not. The reason for this might be that both image patches and their context plays the key role in the object detection task and how the high-order co-occurrence statistics between image patches affect the detection results still remains unclear.

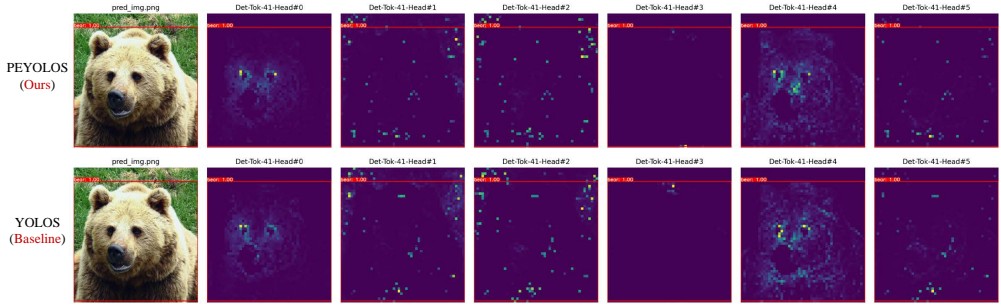

Figure 12: The self-attention map visualization of the detection tokens and the corresponding predictions, of the last layer attention heads based on PEYOLOS (Deit-S) and YOLOS (Deit-S).

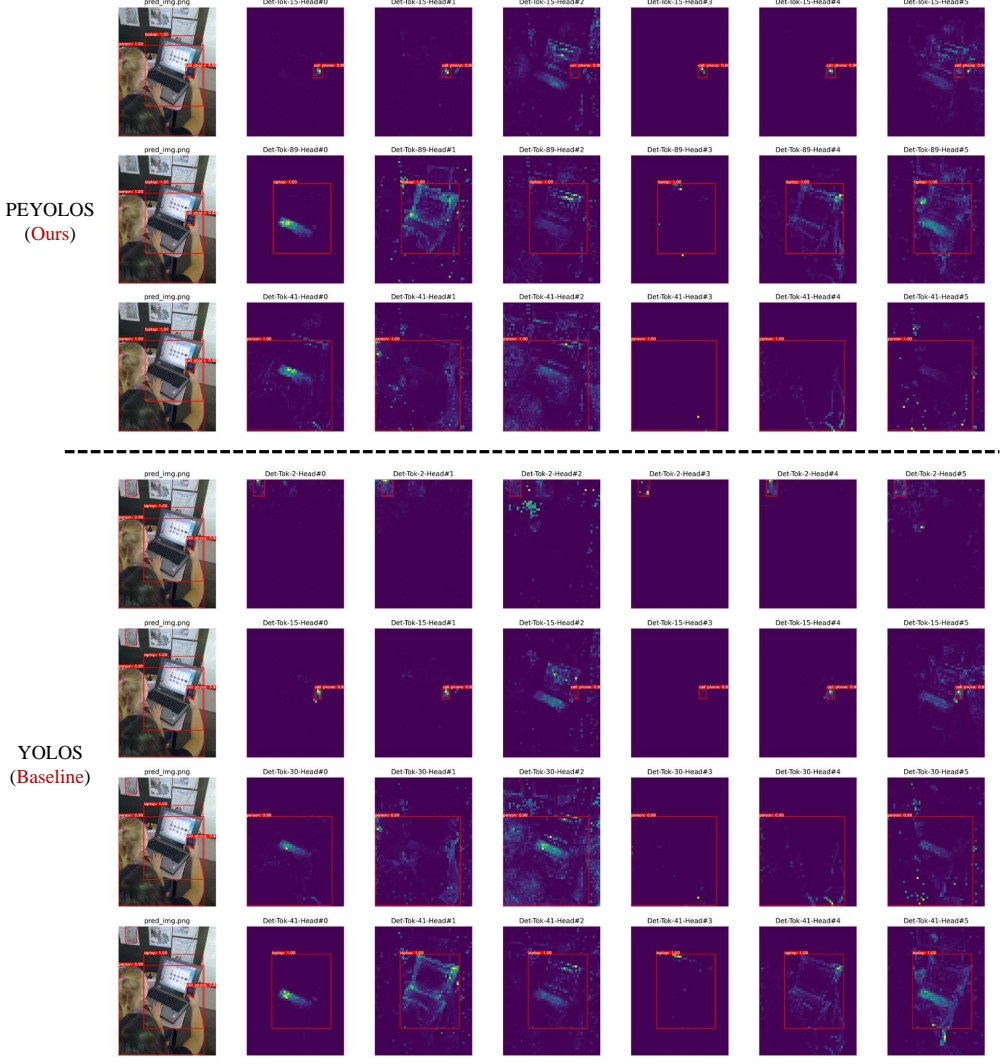

Figure 13: The self-attention map visualization of the detection tokens and the corresponding predictions, of the last layer atttion heads based on PEYOLOS (Deit-S) and YOLOS (Deit-S).

**Ablation on Patch Size for Object Detection.** We have conducted ablation experiments on the patch size for object detection, in which the backbone used is DeiT-Ti. The results are shown in Table 5. It can be observed that the best performance is achieved with the patch size of $16 \times 16$. Therefore, the patch size is set to $16 \times 16$ by default.

Table 5: The impact of patch size on object detection performance. Here, the backbone used is DeiT-Ti.

| Patch size | Accuracy (AP) |
|---|---|
| $20 \times 20$ | 20.7 |
| $16 \times 16$ | 25.3 |
| $14 \times 14$ | 23.4 |
| $10 \times 10$ | 23.2 |

**Impact of nonlinear embedding.** The results of PEYOLOS with linear embedding and without nonlinear embedding are shown in Table 6. It can be observed that nonlinear embedding can boost the performance of PEYOLOS. The reason for this might be that nonlinear embedding has stronger capacity, so it can capture more details useful for object detection.

Table 6: Impact of nonlinear embedding on object detection performance on COCO. Here, the backbone used is DeiT-Ti.

| Method | AP |
|---|---|
| PEYOLOS with linear embedding | 23.9 |
| PEYOLOS with two-layer nonlinear embedding | 25.3 |
| PEYOLOS with three-layer nonlinear embedding | 26.2 |

## D    RECONSTRUCTION ATTACKER

We use a recently proposed powerful Transformer-based framework, MAE He et al. (2022) (tiny), to recover the original clean images from the images encrypted by MI. We adapt MAE with two modifications: (1) Patches are not dropped and (2) The linear patch embedding is replaced by a nonlinear patch embedding, which is consistent with the patch embedding used in PEYOLOS. Here the nonlinear patch embedding is composed of two linear layers separated by a GELU activation. Corresponding results are shown in Figure 14. We observe that: (1) The style of reconstructed images is very different from original images and (2) Privacy-sensitive patches such as faces and texts are blurred, and thus the reconstruction with MAE still cannot reveal the original identity of faces or the contents of texts. These observations indicate that recovering the original clean natural images from images encrypted by MI is a great challenge, demonstrating the effectiveness of MI regarding privacy preserving.

## E    TRANSFORMER-BASED PUZZLE SOLVER ATTACKER

For the large-scale dataset ImageNet, the method in Mena et al. (2018) solves the jigsaw puzzle problem with $3 \times 3$ patches. Even in this simple case, the authors still face big challenges. For example, the authors claimed that "Learning in the Imagenet dataset is much more challenging, as there isnt a sequential structure that generalizes among images".

We extend Mena et al. (2018) to the case where there are $14 \times 14$ patches (image size $224 \times 224$, patch size $16 \times 16$). We use the powerful pipeline of MAE He et al. (2022) with the following modifications: (1) Patches are not dropped, (2) The loss function used is Gumbel-Softmax proposed in Mena et al. (2018), and (3) The positional encoding is removed, which is necessary as the patch orders are randomly shuffled by RS. The results are shown in Figure 15.

It can be observed that predicting the correct patch orders faces big challenges. The reason for this might be that (1) only a tiny fraction of $196! \approx 5 \times 10^{365}$ possible patch orders are sampled, which is difficult to estimate the underlying joint distribution of patch orders and patch content. Note that

sampling a large fraction of $196! \approx 5 \times 10^{365}$ possible patch orders is infeasible in practice. (2) Extremely strong discriminative power are needed. For examples, if we mirror an image, the patch orders should change accordingly. In this case, the input patches are very similar, and a model needs a very strong discriminative power to distinguish them.

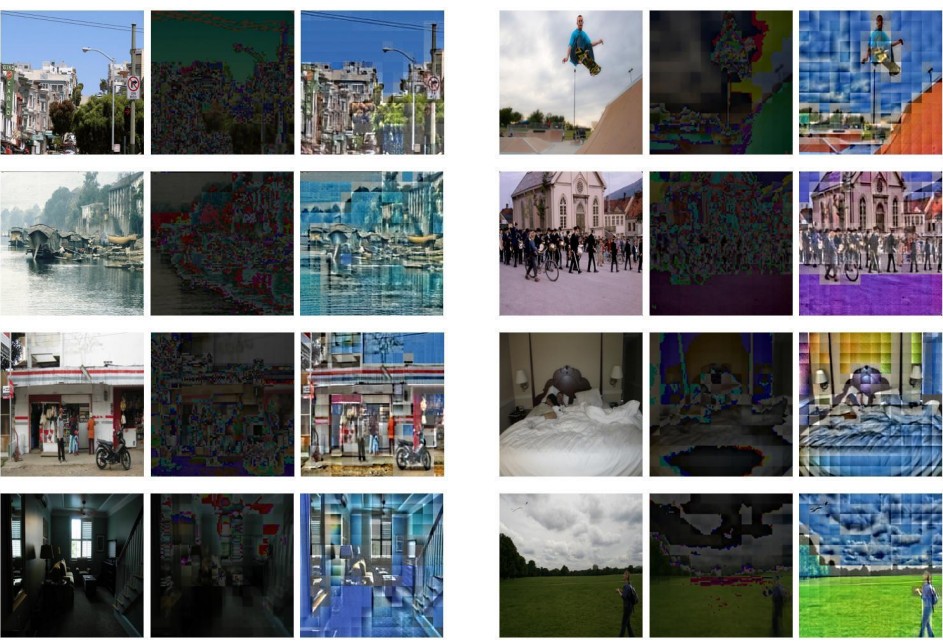

Figure 14: Reconstruction attack on MI. Left: original images. Middle: images encrypted by MI. Right: reconstructed images.

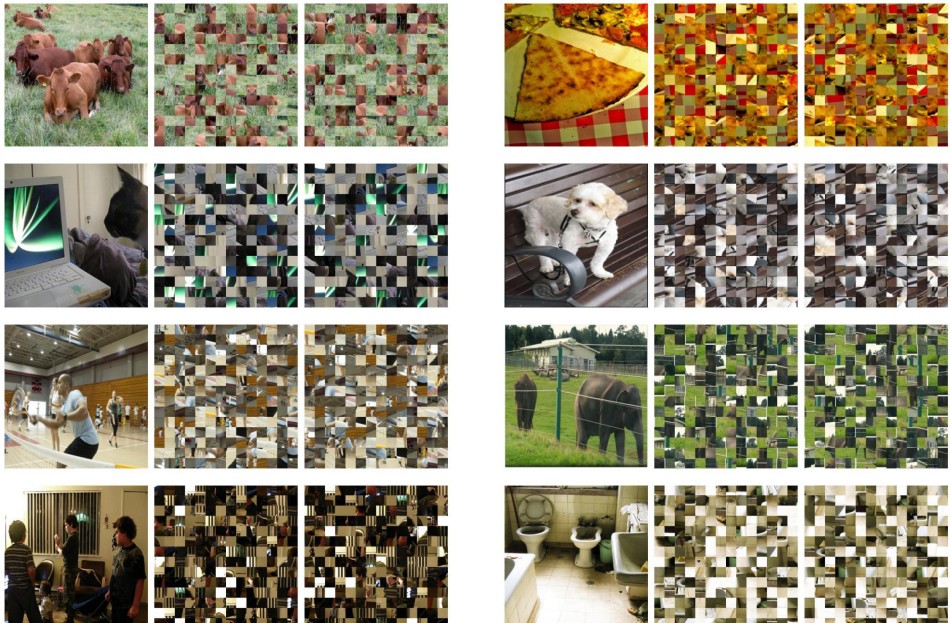

Figure 15: Reconstruction attack on RS. Left: original images. Middle: images encrypted by RS. Right: reconstructed images.

To find the largest number of patches the reconstruction attack works, we finetune a pretrained ViT-base to solve the jigsaw puzzle in the settings of $2 \times 2$, $3 \times 3$, $4 \times 4$, and $5 \times 5$ patches. The training loss converges in the cases of $2 \times 2$ and $3 \times 3$ patches, but could not converge in all other cases. This suggests that the threshold is $3 \times 3$: when the patch number is larger than $3 \times 3$, the reconstruction attack does not work any more (see Figure 16). Our encryption is thus safe from ViT attack, where the patch number is $14 \times 14$.

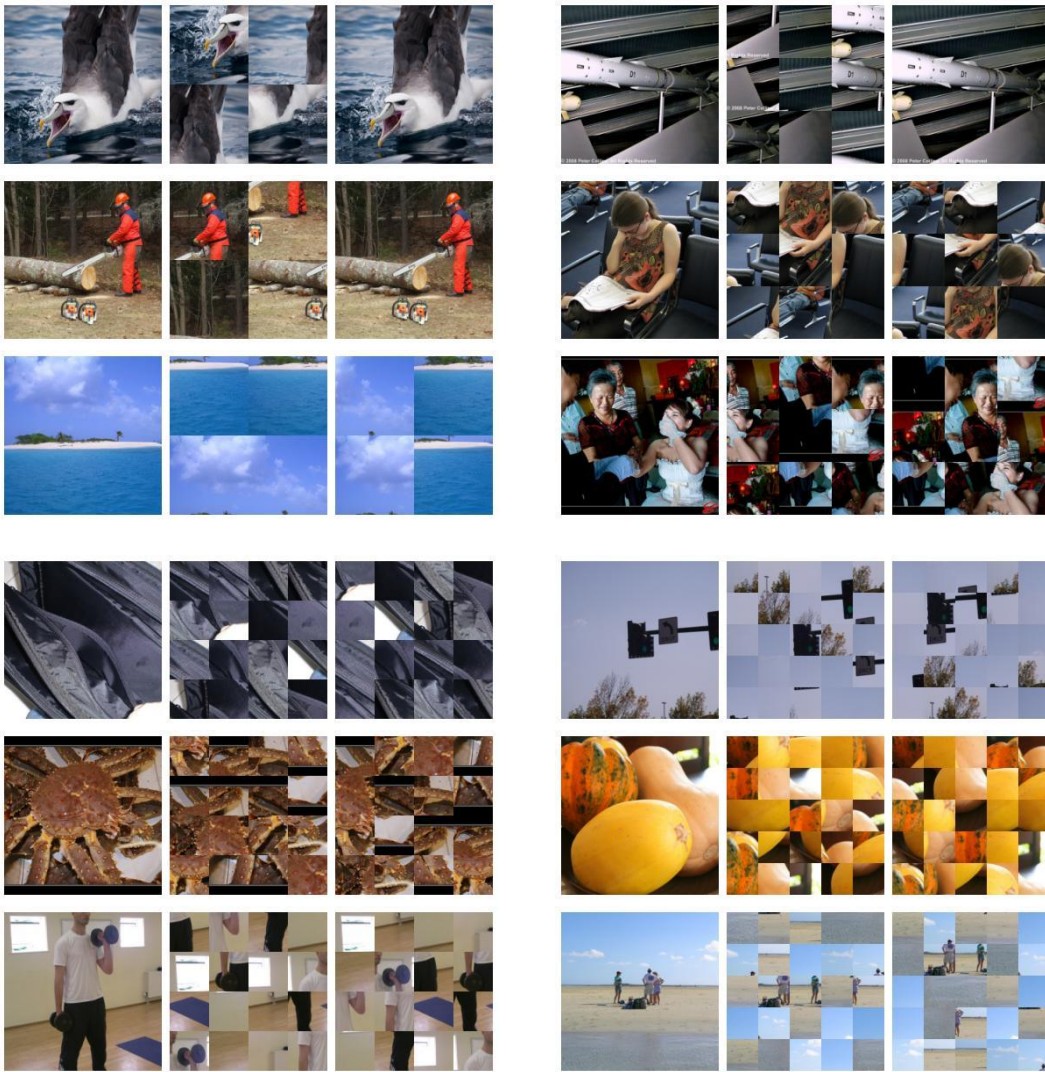

Figure 16: Impact of patch number on the reconstruction attack effect. Left: original images. Middle: permutated images. Right: reconstructed images.

# F  GRADIENT LEAKAGE ATTACKER

It has been shown that the training set will be leaked by gradient sharing Zhu et al. (2019). To evaluate the impact of gradient leakage on the security of our method, we use the gradient leakage attacker to recover the images encrypted by RS. The results are shown in Figure 17. It can be observed that (1) Gradient leakage attacker can restore images and (2) the restored images are encrypted. Therefore, we conclude that our paradigm does not prevent gradient leakage attacks, but can make the attacked images useless, thus protecting privacy.

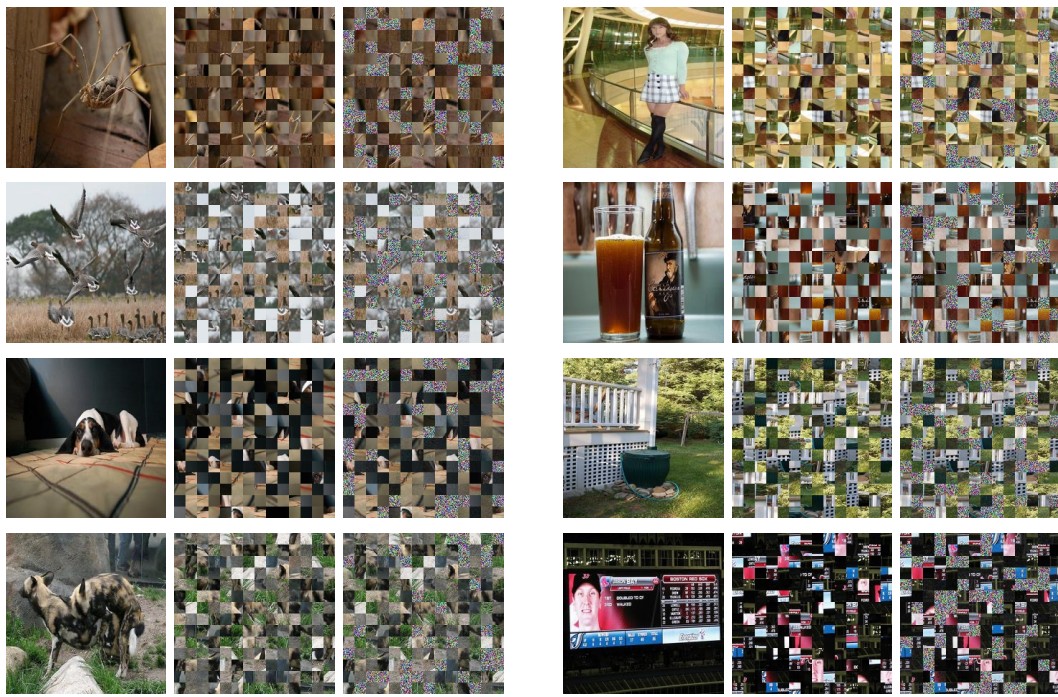

Figure 17: Impact of gradient leakage on the security of our method. Left: original images. Middle: images encrypted by RS. Right: recovered images.

# G  A QUANTITATIVE MEASURE OF PRIVACY LEAKAGE

We design a simple measure to quantify the degree of privacy disclosure i.e., detecting faces on original images and encrypted images and then considering the ratio of faces detected on encrypted images as a measure, in which the face detection method used is MTCNN [2]. We have conducted experiments with this measure, and the results are shown in Table 7. It can be observed that both RS and MI can protect the visual contents to a great extent.

Table 7: Quantitative measure of privacy leakage. Please note that the experiments are conducted on the validation set of ImageNet.

| Image source | # of Detected faces | Ratio of leakage |
|---|---|---|
| Original images | 15242 | - |
| Encrypted image by RS | 1337 | 8.8% |
| Encrypted image by MI | 212 | 1.4% |

We also perform experiments to measure the degree of privacy preservation by asking forty people to evaluate the effect of encryption and attack on 100 images (image size $224 \times 224$) and the results are shown in Table 8. It can be observed that (1) Both RS and MI can protect privacy well in most

---

[2]https://github.com/timesler/facenet-pytorch

cases and (2) The combination of RS and MI is hard to decrypt. In summary, we would like to stress out that the larger the number of patches, the higher the security level and the overall encryption strength can be further improved by dropping sensitive patches, integrating MI, etc.

Table 8: Degree of privacy preservation evaluated by human beings.

| Method | Able to recognize image content beyond the label | Hard to recognize image content beyond the label |
|---|---|---|
| Images encrypted by RS (patch size $16 \times 16$) | 0% | 100% |
| Images encrypted by RS (patch size $8 \times 8$) | 0% | 100% |
| Images encrypted by MI | 2% | 98% |
| Images encrypted by RS + MI | 0% | 100% |
| Puzzle solver attack on RS (patch size $16 \times 16$) Sec. 4.4 | 10% | 90% |
| Puzzle solver attack on RS (patch size $8 \times 8$) Sec. 4.4 | 3% | 97% |
| Reconstruction attack on MI Sec. D in the appendix | 6% | 94% |
| Puzzle solver and reconstruction attack on RS + MI | 0% | 100% |

## H IMPLEMENTATION DETAILS OF RS AND MI

The key implementation details of RS and MI are provided in Algorithm 1 and Algorithm 2.

---

**Algorithm 1** Pseudocode of RS in a PyTorch-like style.

---

```
class RS_PatchEmbed(nn.Module):
# Key implementation details of RS
def __init__(self, patch_size, in_chans, embed_dim):
super().__init__()
patch_size = to_2tuple(patch_size)
self.proj = nn.Conv2d(in_chans, embed_dim, kernel_size=patch_size, stride=
    patch_size)

def forward(self, x):
B, C, H, W = x.shape
x = self.proj(x).flatten(2).transpose(1, 2) # B x HW x C

shuffle = []
for idx in range(B):
random_idx = torch.randperm(x.size(1))
shuffle.append(x[idx][random_idx, :].unsqueeze(0))

x = torch.cat(shuffle, dim=0)

return x
```

---

**Algorithm 2** Pseudocode of MI in a PyTorch-like style.

---

```
class MI_PatchEmbed(nn.Module):
# Key implementation details of MI
# MI is implemented based on the fact that $0.25*W\sum_{i=1}^{4}{x_i}=0.25*\sum_
    {i=1}^{4}{Wx_i}$

def __init__(self, patch_size, in_chans, embed_dim):
super().__init__()
patch_size = to_2tuple(patch_size)
self.proj_1 = nn.Conv2d(in_chans, embed_dim, kernel_size=patch_size,stride=
    patch_size)
self.pool = nn.AvgPool2d(2, stride=2)
self.act = nn.GELU()
self.proj_2 = nn.Conv2d(embed_dim, embed_dim, kernel_size=1, stride=1)

def forward(self, x):
B, C, H, W = x.shape
x = self.proj_1(x) # B x C x (H/8) x (W/8)
x = self.pool(x) # B x C x (H/16) x (W/16)
x = self.proj_2(self.act(x)).flatten(2).transpose(1, 2) # B x HW x C

return x
```

---

