# OpenReview forum: "Privacy-Preserving Vision Transformer on Permutation-Encrypted Images"
_ICLR.cc/2023/Conference — Submitted to ICLR 2023_

### Official Review · Reviewer_ZYXS · 2022-10-24

**Confidence:** 4
**Correctness:** 3
**Technical Novelty And Significance:** 3
**Empirical Novelty And Significance:** 3
**Recommendation:** 5

**Clarity, Quality, Novelty And Reproducibility:**

Clarity: the paper is easy to follow; the adaptation of vision transformers is straightforward.

Quality and Novelty: Somewhat limited because the degree of privacy preservation cannot be measured and compared quantitatively. Also, the use MI as a non-invertible operation is arguable.

Reproducibility: the paper's results are easily reproducible provided the seeds of random shuffling are given.

**Strength And Weaknesses:**

Strengths:

1. Topic is important.
2. Result is promising.

Weaknesses:

1. The primary issue is the lack of quantitative metrics to measure the degree of privacy preservation. Visual inspection of Figure 5 indicates that the main content has already been revealed. Although the authors emphasize that solving the jigsaw puzzle problem is NP-hard, in practice, we only need to find a not-so-good local optimal (e.g., in the mean squared error sense) that is sufficient to reveals the main semantics w.r.t. the task at hand.

2. If the reviewer understands correctly, after the MI encryption process, even the encrypters themselves cannot perfectly recover the original image (see Eq. (2)), which is unfavorable. We expect encryption and the corresponding decryption together to form a bijection.

3. For image classification, we tend to rely on permutation-equivariant and permutation-invariant operations as suggested by the authors. Then, what is the motivation to incorporate a reference-based positional embedding in Eq. (10)?

4. The math notation in Eq. (12) is less clear. Specifically, what is $\mathbf{x}_i^\mathrm{DET}$?

**Summary Of The Paper:**

This paper tackles privacy-preserving image classification and object detection (in the context of vision transformers) based on random shuffling and mixing-up on the patch scale.


**Summary Of The Review:**

With a much more rigorous privacy-preserving analysis and the justification of the MI encryption, the paper has a chance to get in.

---

> ### Author Response · Authors · 2022-11-18
> **Response to Reviewer ZYXS (Part 2/2)**
>
> **Q4.2:** _If the reviewer understands correctly, after the MI encryption process, even the encrypters themselves cannot perfectly recover the original image (see Eq. (2)), which is unfavorable. We expect encryption and the corresponding decryption together to form a bijection._
>
> **A4.2:** Thanks. Firstly, we would like to stress out the background of our privacy protection strategies. Training deep models requires a large number of images and directly collecting original images may reveal information beyond the label. Moreover, due to the high consumption of computing resources, the trained deep models are usually deployed in the clouds and users are required to share their original images with the cloud service providers to get benefited by the on-demand services. In this case, the risk of privacy leakage beyond the label still exists.
>
> To hide all information contained in original images except for their labels, a promising direction is to learn and infer on encrypted images. In this work, we propose an efficient privacy-preserving learning paradigm that can destroy human-recognizable contents while preserving machine-learnable information.
>
> In the above application scenario, users already have original images in hands, and their concern is the possible privacy leakage when releasing original images. In this case, the encryption strength of original images and the performance of models learned on encrypted images may be more concerned by users.
>
> **Q4.3:** _For image classification, we tend to rely on permutation-equivariant and permutation-invariant operations as suggested by the authors. Then, what is the motivation to incorporate a reference-based positional embedding in Eq. (10)?_
>
> **A4.3:** Thanks. The results of PEViT with and without RPE are shown below.
>
> |Method|Top-1 acc.|
> |:---|:---:|
> | PEViT without RPE |78.7%|
> | PEViT with RPE |79.7%|
>
> It can be observed that RPE can boost the performance of PEViT. The motivation for designing RPE is as follows. In PEViT, the positional encoding is removed due to its non-permutation-equivariant property. Through RPE, we illustrate that introducing positional embedding while retaining permutation-equivariant property is feasible, which might inspire other better designs of positional embedding that is permutation-equivariant.
>
> **Q4.4:** _The math notation in Eq. (12) is less clear. Specifically, what is $x_i^{DET}$?_
>
> **A4.4:** Thanks. $x_i^{DET}$ denotes the detection token, similar to class token.
>
>
>
> **Q4.5:** _With a much more rigorous privacy-preserving analysis and the justification of the MI encryption, the paper has a chance to get in._
>
> **A4.5:** Thank you for thorough reviews and constructive suggestions, which inspire us a lot.

---

> ### Author Response · Authors · 2022-11-18
> **Response to Reviewer ZYXS (Part 1/2)**
>
> **Q4.1:** _The primary issue is the lack of quantitative metrics to measure the degree of privacy preservation. Visual inspection of Figure 5 indicates that the main content has already been revealed. Although the authors emphasize that solving the jigsaw puzzle problem is NP-hard, in practice, we only need to find a not-so-good local optimal (e.g., in the mean squared error sense) that is sufficient to reveals the main semantics w.r.t. the task at hand._
>
> **A4.1:** Thanks. Jigsaw puzzle solving is an NP-hard problem [a]. This implies that, likely, no efficient algorithm can solve jigsaw puzzle in general. At present, all known algorithms for NP-complete problems require time that is super-polynomial in the input size, in fact exponential in $O(n^k)$ for some $k>0$, and it is unknown whether there are any faster algorithms.
>
> We have tried to reconstruct images encrypted by RS with different puzzle solver attackers, including the Transformer-based attacker; please see Sec. 4.4 and Sec. E in the appendix. It is found that for the typical value of the number of patches (14x14 or 28x28) used by RS, these attackers cannot reconstruct the original images. Since the order of patches cannot be accurately predicted, their attack success rate is 0. Considering that privacy would be leaked if some local regions containing sensitive information could be reconstructed, the above metric is not accurate. Therefore, we do not report this quantitative indicator.
>
> We agree with the reviewer that a not-so-good local optimal may be sufficient to reveal the main semantics. Fortunately, this concern can be addressed by the following two ways. (1) It is found that even if 10% of the patches are dropped, the performance is only reduced by 1.6%. This allows users to drop patches containing sensitive information at the cost of slight performance degradation to further enhance the encryption strength of RS. Please see Sec. 4.4 for more details. (2) The encryption strength of RS can be further enhanced by integrating MI, where MI makes the image content difficult for human eyes to distinguish. Please see Figure 1.
>
> We have performed experiments to measure the degree of privacy preservation. Specifically, forty people were asked to evaluate the effect of encryption and attack on 100 images (image size 224 x 224) and the results are shown below.
>
> |Method| Able to recognize image content beyond the label | Hard to recognize image content beyond the label |
> |:---|:---:|:---:|
> |Images encrypted by RS (patch size 16 x 16) | 0% | 100% |
> |Images encrypted by RS (patch size 8 x 8) | 0% | 100% |
> |Images encrypted by MI | 2% | 98% |
> |Images encrypted by RS + MI | 0% | 100% |
> |Puzzle solver attack on RS (patch size 16 x 16); Sec. 4.4 | 10% | 90% |
> |Puzzle solver attack on RS (patch size 8 x 8); Sec. 4.4 | 3% | 97% |
> |Reconstruction attack on MI; Sec. D in the appendix | 6% | 94% |
> |Puzzle solver and reconstruction attack on RS + MI | 0% | 100% |
>
> It can be observed that (1) Both RS and MI can protect privacy well in most cases and (2) The combination of RS and MI is hard to decrypt.
>
> In summary, *the larger the number of patches, the higher the security level* and *the overall encryption strength can be further improved by dropping sensitive patches, integrating MI, etc.*
>
> [a] E. Demaine and M. Demaine. Jigsaw puzzles, edge matching, and polyomino packing: Connections and complexity. Graphs and Combinatorics, 23:195–208, 2007.

---

### Official Review · Reviewer_E1ZJ · 2022-10-24

**Confidence:** 5
**Correctness:** 1
**Technical Novelty And Significance:** 1
**Empirical Novelty And Significance:** Not applicable
**Recommendation:** 1

**Clarity, Quality, Novelty And Reproducibility:**

Clarity:
- paper is well-written and easy to understand

Reproducibility:
- authors provide a source code in supplemental material, which should make it easy to reproduce the result. As a note, I haven't actually inspected or run the code.

Novelty:
- I think the fact that vision transformer does not really need positional embedding has some amount of novelty. However I'm not sure to what degree this topic is already explored in other papers, thus amount of novelty might be quite limited and do not justify for a conference paper. In particular [Shaw et al 2018] work which is referenced in the paper already exploring relative positional embedding, instead of absolute embedding.

Quality:
- As mentioned in the weaknesses section, privacy and encryption claims of the paper are essentially invalid.



**Strength And Weaknesses:**

Strength:
1) Paper is reasonably well-written and easy to understand.
2) It is an interesting observation that vision transformer does not need positional embedding for successful image recognition. Idea of RLE is also interesting.

Weaknesses:
1) Authors claim "encryption" and "privacy protection", however they don't define what they mean by privacy and don't define their threat model. One of the adopted notions of privacy for ML models is differential privacy ( https://arxiv.org/abs/1607.00133 ). It's fine to define and use different notion of privacy, but it has to be well defined. It seems like, by privacy authors imply inability for a human observer to recognize what's on the image. However this is highly subjective, moreover proposed methods arguable don't even fulfill this goal (see below).
2) Lacking evaluation of reconstruction attack. I didn't find any numerical evaluation. Evaluation in experimental section is summarized by a quote: "As shown in Figure 1, the visual contents of encrypted images are nearly-completely protected from recognizing by human eyes". For the reference, figure 1 shows a few images before and after encoding. So it appear to me that authors claim that solving jigwas puzzle is a hard problem, showing few images before and after encoding and claim that this is the reason why their method protects privacy.
3) Authors over estimate complexity of solving a jigsaw puzzle. The experiments are done in a setting when RS method split image into 14x14 patches. Authors refer to a paper about automated jigsaw puzzle solver from 2015 and claim that it's a sufficiently hard problem.
Arguable, individual 14x14 jigsaw puzzle (196 pieces) could be easily solved by human in a few hours at most. Which means that any individual image encoded with RS method does not really "destroy human recognizable content".
For comparison, no human can manually decrypt a file, which is encrypted with AES ( https://en.wikipedia.org/wiki/Advanced_Encryption_Standard )


**Summary Of The Paper:**

This paper proposes to "encrypt" images by using two techniques. First one (RS) is to cut each image into patches and then reshuffle the order of tiles. Second one (MI) is to mix up some of the patches.
Authors show that vision transformer without positional embeddings (PEViT) can successfully learn to recognize images encoded by RS method. And they also show a modification of object detector YOLO which can perform object detection on images encoded by MI method.

**Summary Of The Review:**

Strong reject.

Paper claim to propose an encryption scheme which protects privacy. However authors neither follow any common privacy definition (for example, differential privacy), nor they define their own notion of privacy. In addition, evaluation of encryption properties of the proposed methods is lacking.

---

> ### Author Response · Authors · 2022-11-18
> **Response to Reviewer E1ZJ (Part 4/4)**
>
> **Q3.4:** _I think the fact that vision transformer does not really need positional embedding has some amount of novelty. However I'm not sure to what degree this topic is already explored in other papers, thus amount of novelty might be quite limited and do not justify for a conference paper. In particular [Shaw et al 2018] work which is referenced in the paper already exploring relative positional embedding, instead of absolute embedding._
>
> **A3.4:** Thanks. Training deep models requires a large number of images and directly collecting original images may reveal information beyond the label. Moreover, due to the high consumption of computing resources, the trained deep models are usually deployed in the clouds and users are required to share their original images with the cloud service providers to get benefited by the on-demand services. In this case, the risk of privacy leakage beyond the label still exists.
>
> To avoid privacy leakage beyond the label, we propose an efficient privacy-preserving learning paradigm that can destroy human-recognizable contents while preserving machine-learnable information. Extensive experiments on ImageNet and COCO show that our proposed paradigm achieves comparable accuracy with the competitive methods while preserving privacy.
>
> Moreover, we would like to stress out that the relative positional embedding is **not permutation-equivariant**. By contrast, our reference-based positional embedding is permutation-equivariant. Through RPE, we illustrate that introducing positional embedding while retaining permutation-equivariant property is feasible, which might inspire other better designs of positional embedding that is permutation-equivariant.

---

> ### Author Response · Authors · 2022-11-18
> **Response to Reviewer E1ZJ (Part 3/4)**
>
> **Q3.3:** _Authors over estimate complexity of solving a jigsaw puzzle. The experiments are done in a setting when RS method split image into 14x14 patches. Authors refer to a paper about automated jigsaw puzzle solver from 2015 and claim that it's a sufficiently hard problem. Arguable, individual 14x14 jigsaw puzzle (196 pieces) could be easily solved by human in a few hours at most. Which means that any individual image encoded with RS method does not really "destroy human recognizable content". For comparison, no human can manually decrypt a file, which is encrypted with AES ( https://en.wikipedia.org/wiki/Advanced_Encryption_Standard )_
>
> **A3.3:** Thanks. Firstly, we would like to stress out that jigsaw puzzle solving is an NP-complete problem [b]. This implies that, likely, no efficient algorithm can solve jigsaw puzzle in general. At present, all known algorithms for NP-complete problems require time that is super-polynomial in the input size, in fact exponential in $O(n^k)$ for some $k>0$, and it is unknown whether there are any faster algorithms.
>
> Considering that the complexity of solving jigsaw puzzle increases rapidly with the number of patches, the encryption strength of RS is closely related to the number of patches. In other words, *the number of patches determines security levels, and the larger the number of patches, the higher the security level*. This is very similar to the definition of the security level of a shredder, that is, the security level is determined by the size of the paper shreds.
>
> To investigate the security level of RS, we have finetuned the powerful ViT to solve the jigsaw puzzle with $2\times2$, $3\times3$, $4\times4$, or $5\times5$ patches. The training losses of jigsaw puzzles with $2\times2$ and $3\times3$ patches converge, while the others could not converge. This shows that the largest number of patches the ViT-based solver works is $3\times3$, which is much lower than the typical value in our work (e.g., $14\times 14$ or $28\times 28$), thus demonstrating the effectiveness of RS. Please see Sec. E in the appendix for more details.
>
> Moreover, it is found that even if 10% of the patches are dropped, the performance is only reduced by 1.6%. This allows users to drop patches containing sensitive information at the cost of slight performance degradation to further enhance the encryption strength of RS. Please see Sec. 4.4 for more details.
>
> It's worth noting that MI can also be integrated into RS to further enhance the encryption strength, where MI makes the image content difficult for human eyes to distinguish. Please see Figure 1.
>
> We have performed experiments to measure the degree of privacy preservation. Specifically, forty people were asked to evaluate the effect of encryption and attack on 100 images (image size 224 x 224) and the results are shown below.
>
> |Method| Able to recognize image content beyond the label | Hard to recognize image content beyond the label |
> |:---|:---:|:---:|
> |Images encrypted by RS (patch size 16 x 16) | 0% | 100% |
> |Images encrypted by RS (patch size 8 x 8) | 0% | 100% |
> |Images encrypted by MI | 2% | 98% |
> |Images encrypted by RS + MI | 0% | 100% |
> |Puzzle solver attack on RS (patch size 16 x 16); Sec. 4.4 | 10% | 90% |
> |Puzzle solver attack on RS (patch size 8 x 8); Sec. 4.4 | 3% | 97% |
> |Reconstruction attack on MI; Sec. D in the appendix | 6% | 94% |
> |Puzzle solver and reconstruction attack on RS + MI | 0% | 100% |
>
> It can be observed that (1) Both RS and MI can protect privacy well in most cases and (2) The combination of RS and MI is hard to decrypt.
>
> We would like to stress out that *the larger the number of patches, the higher the security level* and *the overall encryption strength can be further improved by dropping sensitive patches, integrating MI, etc.*
>
> Existing cryptographic methods such as homomorphic encryption and AES can be used to encrypt original images. Unfortunately, learning on images encrypted by cryptographic methods is a huge challenge due to their high computational overheads, their large performance degradation, or their needs for special setups (e.g., finite field arithmetic, public-key infrastructure) [a].
>
> [a] Yangsibo Huang, Zhao Song, Kai Li, and Sanjeev Arora. InstaHide: Instance-hiding Schemes for Private Distributed Learning, ICML 2020.
> [b] E. Demaine and M. Demaine. Jigsaw puzzles, edge matching, and polyomino packing: Connections and complexity. Graphs and Combinatorics, 23:195–208, 2007.
> [c] Nicholas Carlini, Sanjam Garg, Somesh Jha, Saeed Mahloujifar, Mohammad Mahmoody, and Florian Tramer. NeuraCrypt is not private, arXiv:2108.07256

---

> ### Author Response · Authors · 2022-11-19
> **Response to Reviewer E1ZJ (Part 2/4)**
>
> **Q3.2:** _Lacking evaluation of reconstruction attack. I didn't find any numerical evaluation. Evaluation in experimental section is summarized by a quote: "As shown in Figure 1, the visual contents of encrypted images are nearly-completely protected from recognizing by human eyes". For the reference, figure 1 shows a few images before and after encoding. So it appear to me that authors claim that solving jigwas puzzle is a hard problem, showing few images before and after encoding and claim that this is the reason why their method protects privacy._
>
> **A3.2:** Thanks. We have tried to reconstruct images encrypted by RS with different puzzle solver attackers, including the Transformer-based attacker; please see Sec. 4.4 and Sec. E in the appendix. It is found that for the typical value of the number of patches (14x14 or 28x28) used by RS, these attackers cannot reconstruct the original images. Since the order of patches cannot be accurately predicted, their attack success rate is 0. Considering that privacy would be leaked if some local regions containing sensitive information could be reconstructed, the above metric is not accurate. Therefore, we do not report this quantitative indicator.
>
> To provide a numerical evaluation, we have performed experiments to measure the degree of privacy preservation. Specifically, forty people were asked to evaluate the effect of encryption and attack on 100 images (image size 224 x 224) and the results are shown below.
>
> |Method| Able to recognize image content beyond the label | Hard to recognize image content beyond the label |
> |:---|:---:|:---:|
> |Images encrypted by RS (patch size 16 x 16) | 0% | 100% |
> |Images encrypted by RS (patch size 8 x 8) | 0% | 100% |
> |Images encrypted by MI | 2% | 98% |
> |Images encrypted by RS + MI | 0% | 100% |
> |Puzzle solver attack on RS (patch size 16 x 16); Sec. 4.4 | 10% | 90% |
> |Puzzle solver attack on RS (patch size 8 x 8); Sec. 4.4 | 3% | 97% |
> |Reconstruction attack on MI; Sec. D in the appendix | 6% | 94% |
> |Puzzle solver and reconstruction attack on RS + MI | 0% | 100% |
>
> It can be observed that (1) Both RS and MI can protect privacy well in most cases and (2) The combination of RS and MI is hard to decrypt.
>
> We would like to stress out that *the larger the number of patches, the higher the security level* and *the overall encryption strength can be further improved by dropping sensitive patches, integrating MI, etc.*

---

> ### Author Response · Authors · 2022-11-19
> **Response to Reviewer E1ZJ (Part 1/4)**
>
> **Q3.1:** _Authors claim "encryption" and "privacy protection", however they don't define what they mean by privacy and don't define their threat model. One of the adopted notions of privacy for ML models is differential privacy ( https://arxiv.org/abs/1607.00133 ). It's fine to define and use different notion of privacy, but it has to be well defined. It seems like, by privacy authors imply inability for a human observer to recognize what's on the image. However this is highly subjective, moreover proposed methods arguable don't even fulfill this goal (see below)._
>
> **A3.1:** Thanks. Firstly, we would like to stress out the background of our privacy protection strategies. Training deep models requires a large number of images and directly collecting original images may reveal information beyond the label. Moreover, due to the high consumption of computing resources, the trained deep models are usually deployed in the clouds and users are required to share their original images with the cloud service providers to get benefited by the on-demand services. In this case, the risk of privacy leakage beyond the label still exists.
>
> Based on the above observations, the privacy to be protected in our work can be concretely defined as follows: hiding all information contained in original image except for its label. This is different from differential privacy, which limits the information that attackers can learn about training data. Similar idea is also mentioned in [a] but the scheme has proved to be not private in [c].
>
> Secondly, we would like to stress out that jigsaw puzzle solving is an NP-complete problem [b]. This implies that, likely, no efficient algorithm can solve jigsaw puzzle in general. At present, all known algorithms for NP-complete problems require time that is super-polynomial in the input size, in fact exponential in $O(n^k)$ for some $k>0$, and it is unknown whether there are any faster algorithms.
>
> Considering that the complexity of solving jigsaw puzzle increases rapidly with the number of patches, the encryption strength of RS is closely related to the number of patches. In other words, *the number of patches determines security levels, and the larger the number of patches, the higher the security level*. This is very similar to the definition of the security level of a shredder, that is, the security level is determined by the size of the paper shreds.
>
> Moreover, it is found that even if 10% of the patches are dropped, the performance is only reduced by 1.6%. This allows users to drop patches containing sensitive information at the cost of slight performance degradation to further enhance the encryption strength of RS. Please see Sec. 4.4 for more details.
>
> It's worth noting that MI can also be integrated into RS to further enhance the encryption strength, where MI makes the image content difficult for human eyes to distinguish. Please see Figure 1.
>
> We have performed experiments to measure the degree of privacy preservation. Specifically, forty people were asked to evaluate the effect of encryption and attack on 100 images (image size 224 x 224) and the results are shown below.
>
> |Method| Able to recognize image content beyond the label | Hard to recognize image content beyond the label |
> |:---|:---:|:---:|
> |Images encrypted by RS (patch size 16 x 16) | 0% | 100% |
> |Images encrypted by RS (patch size 8 x 8) | 0% | 100% |
> |Images encrypted by MI | 2% | 98% |
> |Images encrypted by RS + MI | 0% | 100% |
> |Puzzle solver attack on RS (patch size 16 x 16); Sec. 4.4 | 10% | 90% |
> |Puzzle solver attack on RS (patch size 8 x 8); Sec. 4.4 | 3% | 97% |
> |Reconstruction attack on MI; Sec. D in the appendix | 6% | 94% |
> |Puzzle solver and reconstruction attack on RS + MI | 0% | 100% |
>
> It can be observed that (1) Both RS and MI can protect privacy well in most cases and (2) The combination of RS and MI is hard to decrypt.
>
> In summary, we would like to stress out that *the larger the number of patches, the higher the security level* and *the overall encryption strength can be further improved by dropping sensitive patches, integrating MI, etc.*
>
> [a] Yangsibo Huang, Zhao Song, Kai Li, and Sanjeev Arora. InstaHide: Instance-hiding Schemes for Private Distributed Learning, ICML 2020.
> [b] E. Demaine and M. Demaine. Jigsaw puzzles, edge matching, and polyomino packing: Connections and complexity. Graphs and Combinatorics, 23:195–208, 2007.
> [c] Nicholas Carlini, Sanjam Garg, Somesh Jha, Saeed Mahloujifar, Mohammad Mahmoody, and Florian Tramer. NeuraCrypt is not private, arXiv:2108.07256

---

> > ### Comment · Reviewer_E1ZJ · 2022-11-22
> > **Reply to authors and clarification about definition of privacy.**
> >
> > Thanks for your reply. It still does not address my major concerns.
> >
> > One of the major concerns raised by multiple reviewers is lack of definition of what authors imply by privacy protection.
> > Your reply has an attempt to clarify it by saying: "hiding all information contained in original image except for its label". However that's not really a strict measurable definition. "hiding" could mean a lot of things and need to be clarified.
> >
> > I would try to rephrase my concern differently to make it more clear. Let's say you have an input example `(Image_{input}, Label)` and you try to produce a new example `(Image_{hidden}, Label)` using your algorimth.
> > Then I would like to see answers to two following questions:
> > * a) what procedure or criteria can you use to tell that `Image_{hidden}` is actually protecting privacy of `Image_{input}`
> > * b) why the procedure or criteria you described in a) is useful and worth considering
> > To address b) it would be particularly useful for explain security/privacy game you are considering. In other words, what potential adversary trying to achieve, what adversary can observe and do.
> >
> > None of these is defined in the paper or in the reply.
> >
> > As an illustration of a good privacy definition I would use differential privacy (DP).
> > DP has formal mathematical definition which answers a).
> > Let's say this mathematical definition is satisfied for some algorithm with sufficiently small epsilon. Then such algorithm would be essentially "blind" to exclusion or inclusion of any example into training dataset. This means that an adversary who can only observe an output of such algorithm can not tell anything useful about any individual example in the input dataset.

---

> > > ### Comment · Reviewer_Y75t · 2022-11-22
> > > **Reply to authors**
> > >
> > > I share E1ZJ's concerns regarding the lack of a clear definition of privacy, and the need for questions a) and b) to be addressed.

---

> > > > ### Author Response · Authors · 2022-11-28
> > > > **Follow-up response to Reviewer Y75t**
> > > >
> > > > We appreciate the reviewer for constructive feedback!
> > > >
> > > > Please see our latest response titled definition of privacy. Thanks!

---

> > > ### Author Response · Authors · 2022-11-28
> > > **Definition of privacy**
> > >
> > > We appreciate the reviewer for constructive feedback!
> > >
> > > We first would like to introduce the application scenarios we are considering in a client-server system. Suppose a client hopes the server to train a machine learning model. This client wishes to share training data pairs $(image, label)$ with the server, and then receives the prediction labels of test data $image_{novel}$ from server, but does not want to disclose any information beyond the label. The server collects the data pairs from clients, learns on the collected data, and provides prediction services for clients. To address clients’ privacy concerns, directly learning on encrypted images is a promising solution, while the client needs to encrypt the original, clear data $image$ into $image_{hidden}$. In this way, both the server and potential adversaries cannot access information beyond the label. This paradigm protects sensitive data during data sharing.
> > >
> > > Secondly, we would like to recall a framework termed *substitution-permutation network* (SPN) in cryptography [b]. SPN is a series of linked mathematical operations used in block cipher algorithms such as AES and DES [b]. Shannon suggests that practical and secure ciphers may be constructed by employing a mixing transformation consisting of several rounds of *confusion and diffusion* [a]; SPN is exactly an implementation of this *confusion and diffusion* paradigm. Such an implementation applies several alternating rounds of substitution boxes (S-boxes) and permutation boxes (P-boxes) to produce the ciphertext. An S-box substitutes a small block of bits (the input of the S-box) by another block of bits (the output of the S-box). A P-box is a permutation of bit blocks (or bits). Although a single typical S-box or a single P-box alone does not have sufficient cryptographic strength, a well-designed SPN with several alternating rounds of S-boxes and P-boxes already has a very strong proven security [c].
> > >
> > > Our encryption scheme adheres to SPN, while the basic unit of our encryption scheme is pixels instead of bits. In particular, MIxing-up (MI, which substitutes a patch by the mixup of its sub-patches), can be seen as an implementation of S-box. Random-Shuffling (RS, which permutes the order of patches), can be seen as an implementation of P-box. Although both MI and RS alone does not have sufficient cryptographic strength, alternating several rounds of MI and RS can enhance the cryptographic strength to a large extent. In this way, $image_{hidden}$ is actually protecting privacy of $image$. Hope this can answer a).
> > >
> > > Our encryption scheme is based on a framework with provable security. It is very hard for a potential adversary to steal information from images encrypted by MI + RS (see Figure 1). Further, it is even infeasible with current technologies to infer information from images encrypted by several rounds of MI and RS. We thus conclude that our proposed paradigm is useful and worth considering, which may answer b).
> > >
> > > We will describe this procedure in Sec. 3.
> > >
> > > **In summary, we provide a formal mechanism under the framework of SPN, which verifies that $image_{hidden}$ is exactly protecting privacy of $image$. When this procedure involves sufficient alternating rounds of MI and RS, image contents would be essentially "irrecoverable”. This means that an adversary cannot exploit anything useful about any encrypted images.**
> > >
> > > [a] C.E. Shannon.“Communication Theory of Secrecy Systems,” Bell System Tech. J., 28, 1949, pp. 656-715.
> > >
> > > [b] Douglas R. Stinson and Maura B. Paterson. “Cryptography: theory and practice (Fourth Edition),”Chapman and Hall/CRC, 2019.
> > >
> > > [c] Dodis Yevgeniy, Katz Jonathan, Steinberger John, Thiruvengadam Aishwarya, and Zhang Zhe. “Provable security of substitution-permutation networks,” Cryptology ePrint Archive, 2017.

---

### Official Review · Reviewer_Y75t · 2022-10-25

**Confidence:** 4
**Correctness:** 3
**Technical Novelty And Significance:** 2
**Empirical Novelty And Significance:** 2
**Recommendation:** 5

**Clarity, Quality, Novelty And Reproducibility:**

**Clarity:** The paper is well-organized and clearly written.

**Quality:** The paper is likely to have modest impact within a subfield of AI.

**Novelty:** The paper contributes some new ideas.

**Reproducibility:** Key resources (e.g., proofs, code, data) are available and key details (e.g., proof sketches, experimental setup) are comprehensively described such that competent researchers will be able to easily reproduce the main results.

**Strength And Weaknesses:**

# Strengths

* The main idea of the paper of combining permutation-based image encryption algorithms with transformers is interesting and novel.

* Experimental evaluations on ImageNet and COCO demonstrate that the proposed methods achieve near SOA performance despite operating on encrypted inputs.


# Weaknesses

* The paper fails to provide a concrete definition of privacy. For example, the paper claims that the “proposed paradigm can destroy human-recognizable contents”, but never specifies what is meant by this.

* Given that the stated aim of the proposed encryption algorithms is to “destroy human-recognizable contents”, humans should play some role in the evaluations. Where are the human studies?

* The paper is missing ablation studies for the novel transformer architectures. Where are the experiments demonstrating the need for the proposed RPE in PEViT and the nonlinear embedding in PEYOLOS?

**Summary Of The Paper:**

The paper proposes RS and MI, two permutation-based image encryption algorithms designed to render images unrecognizable to humans, yet still enable image classification and object detection respectively, by downstream vision transformers. The first step of both encryption algorithms is to partition images in to N equally sized patches. Next, RS, short for “random shuffling”, removes the positional embedding of the patches and shuffles their order, leaving a “bag of words” style input. MI, short for “mixing-up”, partitions each of the N patches into M sub-patches and sets each sub-patch equal to the mean of the M sub-patches. The encrypted images are then fed to "permutation-equivariant" variations of ViT and YOLOS, referred to as PEViT and PEYOLOS for image classification and object detection respectively. Compared to ViT, PEViT replaces the conventional positional embedding with a novel reference-based positional embedding. Compared to YOLOS, PEYOLOS replaces the linear patch embedding with a two-layer non-linear patch embedding. Experimental evaluations on ImageNet and COCO demonstrate that the proposed methods achieve near SOA performance despite operating on encrypted inputs.

**Summary Of The Review:**

The main idea, albeit simple, is interesting and novel and seems to produce good results. However, the evaluation is lacking in that privacy is not defined and is missing key ablation studies.

---

> ### Author Response · Authors · 2022-11-18
> **Response to Reviewer Y75t (Part 2/2)**
>
> **Q2.3:** _The paper is missing ablation studies for the novel transformer architectures. Where are the experiments demonstrating the need for the proposed RPE in PEViT and the nonlinear embedding in PEYOLOS?_
>
> **A2.3:** Thanks. The results of PEViT with and without RPE are shown below.
>
> |Method|Top-1 acc.|
> |:---|:---:|
> | PEViT without RPE |78.7%|
> | PEViT with RPE |79.7%|
>
> It can be observed that RPE can boost the performance of PEViT. The motivation for designing RPE is as follows. In PEViT, the positional encoding is removed due to its non-permutation-equivariant property. Through RPE, we illustrate that introducing positional embedding while retaining permutation-equivariant property is feasible, which might inspire other better designs of positional embedding that is permutation-equivariant.
>
> The results of PEYOLOS with linear embedding and without nonlinear embedding are shown below.
>
> |Method| AP|
> |:---|:---:|
> | PEYOLOS with linear embedding |23.9|
> | PEYOLOS with two-layer nonlinear embedding |25.3|
> | PEYOLOS with three-layer nonlinear embedding |26.2|
>
> It can be observed that nonlinear embedding can boost the performance of PEYOLOS. The reason for this might be that nonlinear embedding has stronger capacity, so it can capture more details useful for object detection.

---

> ### Author Response · Authors · 2022-11-18
> **Response to Reviewer Y75t (Part 1/2)**
>
> **Q2.1:** _The paper fails to provide a concrete definition of privacy. For example, the paper claims that the “proposed paradigm can destroy human-recognizable contents”, but never specifies what is meant by this._
>
> **A2.1:** Thanks. Firstly, we would like to stress out the background of our privacy protection strategies. Training deep models requires a large number of images and directly collecting original images may reveal information beyond the label. Moreover, due to the high consumption of computing resources, the trained deep models are usually deployed in the clouds and users are required to share their original images with the cloud service providers to get benefited by the on-demand services. In this case, the risk of privacy leakage beyond the label still exists.
>
> Based on the above observations, the privacy to be protected in our work can be concretely defined as follows: hiding all information contained in original image except for its label. This is different from differential privacy, which limits the information that attackers can learn about training data. Similar idea is also mentioned in [a] but the scheme has proved to be not private in [c].
>
> To maximize the usability of the resulting encryption strategies, the following two requirements need to be satisfied: (1) The performance degradation of the model learned on encrypted images is not significant or acceptable; (2) the human-recognizable content should be largely interfered, while recovering original images is computationally very expensive.
>
> Existing cryptographic methods such as homomorphic encryption and AES can be used to encrypt original images. Unfortunately, learning on images encrypted by cryptographic methods is a huge challenge due to their high computational overheads, their large performance degradation, or their needs for special setups (e.g., finite field arithmetic, public-key infrastructure) [a].
>
> To address the above issues, we design two encryption strategies that can destroy human-recognizable contents while preserving machine-learnable information, where the machine-learnable information refers to the information used to predict the label. Extensive experiments on ImageNet and COCO show that our proposed paradigm achieves comparable accuracy with the competitive methods while preserving privacy.
>
> [a] Yangsibo Huang, Zhao Song, Kai Li, and Sanjeev Arora. InstaHide: Instance-hiding Schemes for Private Distributed Learning, ICML 2020.
> [b] Nicholas Carlini, Sanjam Garg, Somesh Jha, Saeed Mahloujifar, Mohammad Mahmoody, and Florian Tramer. NeuraCrypt is not private, arXiv:2108.07256
>
> **Q2.2:** _Given that the stated aim of the proposed encryption algorithms is to “destroy human-recognizable contents”, humans should play some role in the evaluations. Where are the human studies?_
>
> **A2.2:** Thanks. We have performed experiments as suggested. Specifically, forty people were asked to evaluate the effect of encryption and attack on 100 images (image size 224 x 224) and the results are shown below.
>
> |Method| Able to recognize image content beyond the label | Hard to recognize image content beyond the label |
> |:---|:---:|:---:|
> |Images encrypted by RS (patch size 16 x 16) | 0% | 100% |
> |Images encrypted by RS (patch size 8 x 8) | 0% | 100% |
> |Images encrypted by MI | 2% | 98% |
> |Images encrypted by RS + MI | 0% | 100% |
> |Puzzle solver attack on RS (patch size 16 x 16); Sec. 4.4 | 10% | 90% |
> |Puzzle solver attack on RS (patch size 8 x 8); Sec. 4.4 | 3% | 97% |
> |Reconstruction attack on MI; Sec. D in the appendix | 6% | 94% |
> |Puzzle solver and reconstruction attack on RS + MI | 0% | 100% |
>
> It can be observed that (1) Both RS and MI can protect privacy well in most cases and (2) The combination of RS and MI is hard to decrypt.
>
> In summary, *the larger the number of patches, the higher the security level* and *the overall encryption strength can be further improved by dropping sensitive patches, integrating MI, etc.*

---

### Official Review · Reviewer_rSs2 · 2022-10-25

**Confidence:** 4
**Correctness:** 4
**Technical Novelty And Significance:** 2
**Empirical Novelty And Significance:** Not applicable
**Recommendation:** 5

**Clarity, Quality, Novelty And Reproducibility:**

The overall writing of the paper is good enough.
The overall idea is somewhat novel, however, the network structure is a bit weak in terms of novelty.
Other comments:
- In table 1, the authors should compare with DeiT-B on images encrypted by RS.
- Are the results of COCO object detection compared with PEYOLOS trained on images encrypted by MI? This should be indicated in table 2 as in table 1 for clarity.

**Strength And Weaknesses:**

Strength:
1. The overall paper is well-written and defined. It is easy enough to follow and reproduce.
2. Extensive experimental results to demonstrate the strength of the proposed method.

Weaknesses:
1. Limitation of the paper is not discussed.
2. The change to the existing method so that it can learn from encrypted images is not significant. For example, the change from the existing YOLOS to the proposed PEYOLOS is the input x^p to x_s as in Eq. 12.



**Summary Of The Paper:**

The paper introduces a privacy-preserving training scheme that consists of two parts: encryption strategies based on permutation-equivariance and (partially) permutation-equivariant learnable network.
The contributions of the paper are the two encryption strategies, names Random Shuffling and Mixing, together with the modifications of the existing transformer-based image classification and object detection to learn from encrypted images.

**Summary Of The Review:**

In summary, the paper still has some valuable contributions in terms of encryption strategies. Thus, my recommendation rating for the paper is weak accept.

---

> ### Author Response · Authors · 2022-11-18
> **Response to Reviewer rSs2**
>
> **Q1.1:** _Limitation of the paper is not discussed._
>
> **A1.1:** Thanks. Although jigsaw puzzle solving is an NP-complete problem, once encrypted, some key patches still have a chance to leak privacy. To address this issue, possible future directions include (1) dropping/masking privacy-sensitive patches and (2) using RS in conjunction with other encryption strategies that can protect patch contents, such as MI, in which positional encodings that are permutation-equivariant may be needed.
>
> **Q1.2:** _The change to the existing method so that it can learn from encrypted images is not significant. For example, the change from the existing YOLOS to the proposed PEYOLOS is the input x^p to x_s as in Eq. 12._
>
> **A1.2:** Thanks. In this work, the encryption strategies are decoupled from the models to be trained, as this would allow the encryption of image contents in scenarios where only limited computing resources are available. Then, we focus on making minimal modifications to existing ViTs to enable them to learn on encrypted images. Specifically, for the image classification task, we remove the positional encoding or use the positional encoding that is permutation-equivariant; for the object detection task, we change the way patches are embedded.
>
> We would like to stress out that *since our encryption strategies require fewer changes to existing ViTs, the modification of stronger ViTs to enable them to learn on encrypted images can be made faster and at lower cost.*
>
> **Q1.3:** _The overall writing of the paper is good enough. The overall idea is somewhat novel, however, the network structure is a bit weak in terms of novelty._
>
> **A1.3:** Thanks and addressed. Please see A1.2.
>
> **Q1.4:** _In table 1, the authors should compare with DeiT-B on images encrypted by RS._
>
> **A1.4:** Thanks. We have included the results of DeiT-B on images encrypted by RS in Table 1. For convenience, the newly added comparison result is shown below.
>
> |Method|Top-1 acc.|
> |:---|:---:|
> |DeiT-B on images encrypted by RS |35.9%|
> |PEViT-B on images encrypted by RS|78.7%|
>
> It can be observed that PEViT-B outperforms DeiT-B by a large margin.
>
> **Q1.5:** _Are the results of COCO object detection compared with PEYOLOS trained on images encrypted by MI? This should be indicated in table 2 as in table 1 for clarity._
>
> **A1.5:** Thanks. Yes. We have indicated this in Table 2 as suggested.

---

> > ### Comment · Reviewer_rSs2 · 2022-12-08
> > **RE: Response to Reviewer rSs2**
> >
> > Thank you for your response. I still have concerns about Q1.1 and 1.2.
> >
> > The definition of privacy is not clear. Is it the identity of the person in the photo or the classification of the photo? The authors should define the scope of their works more specific and clearly.
> > Otherwise, this weaken the encryption strategies where we are trying to hide something that we are not clearly know or defined.

---

### Public Comment · ~Nicholas_Carlini1 · 2022-11-04
**This scheme is broken.**

I have broken this scheme. arXiv has been rejecting my break for the past two weeks---with luck they'll change their mind at some point---but I've implemented a jigsaw puzzle solver that can correctly reconstruct images with high probability.

My attack works with three simple steps:
1. I train a neural network to predict if two patches could be adjacent or not
2. I compute the probability any two of the n^2 patches could be placed next to each other
3. I walk the graph to find the best alignment

This allows me to achieve perfect reconstruction for 23% of the ImageNet images I used during testing of my attack, and near-perfect reconstruction of a further 46%.

I do not understand why we should have ever believed that it would be possible to achieve privacy with jigsaw puzzles---a task that is regularly treated as "play" for eight-year-old children.

---

> ### Author Response · Authors · 2022-11-18
> **How many patches are in the jigsaw puzzle you solved?**
>
> Thanks for your interest in our work. Firstly, we would like to stress out that jigsaw puzzle solving is an NP-complete problem [a]. This implies that, likely, no efficient algorithm can solve jigsaw puzzle in general. At present, all known algorithms for NP-complete problems require time that is super-polynomial in the input size, in fact exponential in $O(n^k)$ for some $k>0$, and it is unknown whether there are any faster algorithms.
>
> Considering that the complexity of solving jigsaw puzzle increases rapidly with the number of patches, the encryption strength of RS is closely related to the number of patches. In other words, *the number of patches determines security levels, and the larger the number of patches, the higher the security level*. This is very similar to the definition of the security level of a shredder, that is, the security level is determined by the size of the paper shreds.
>
> To investigate the security level of RS, we have finetuned the powerful ViT to solve the jigsaw puzzle with $2\times2$, $3\times3$, $4\times4$, or $5\times5$ patches. The training losses of jigsaw puzzles with $2\times2$ and $3\times3$ patches converge, while the others could not converge. This shows that the largest number of patches the ViT-based solver works is $3\times3$, which is much lower than the typical value in our work (e.g., $14\times 14$ or $28\times 28$), thus demonstrating the effectiveness of RS. Please see Sec. E in the appendix for more details.
>
> Moreover, it is found that even if 10% of the patches are dropped, the performance is only reduced by 1.6%. This allows users to drop patches containing sensitive information at the cost of slight performance degradation to further enhance the encryption strength of RS. Please see Sec. 4.4 for more details.
>
> It's worth noting that MI can also be integrated into RS to further enhance the encryption strength, where MI makes the image content difficult for human eyes to distinguish. Please see Figure 1.
>
> In summary, **the larger the number of patches, the higher the security level** and **the overall encryption strength can be further improved by dropping sensitive patches, integrating MI, etc.**
>
> Based on the above observations, it would be nice to point out the number of patches in the jigsaw puzzle you solved. Without knowing more details, we respectfully disagree with you that our scheme is broken.
>
> [a] E. Demaine and M. Demaine. Jigsaw puzzles, edge matching, and polyomino packing: Connections and complexity. Graphs and Combinatorics, 23:195–208, 2007.

---

### Decision · Program_Chairs · 2023-01-20

**Decision:**

Reject

**Justification For Why Not Higher Score:**

This paper makes claims on security and privacy that need to be formalized for them to be confirmed. The presented scheme is unlikely to pass generally accepted definitions of security and privacy.

**Justification For Why Not Lower Score:**

N/A.

**Metareview: Summary, Strengths And Weaknesses:**

The paper claims to present a privacy-preserving image-recognition model that “encrypts” images via a jigsaw approach before training. Experiments show that a transformer trained on these “encrypted” images can still accurately recognize the content of the images.

The main shortcoming of this paper is that it does formalize the notion of privacy or encryption in any way. The paper does not specify a threat model, does not describe what notion of cryptographic security the “encryption” is supposed to provide, and does not specify what notion of statistical privacy the presented scheme guarantees. As a result, the claim that the proposed scheme is privacy-preserving is moot.

Indeed, it seems quite unlikely that the scheme satisfies any traditional notions of cryptographic security or statistical privacy. This is underscored by the development of a successful attack against the scheme by one of the commenters. The authors’ defense that this attack may require an attacker to have non-polynomial amounts of compute suggests an implicit threat model that the authors should have made explicit in the paper. Moreover, the particular instantiation of the scheme tested in the paper is apparently insufficient to make it the attacker very difficult. The scheme would likely become harder to break if the authors used more, smaller patches but it is quite likely that then the recognition accuracy would also suffer, which would undermine the central claim of the paper.